# CAM photosynthesis may have conferred an advantage during the Permian–Triassic mass extinction event

Zhen Xu [1,2] ✉, Jason Hilton [3], Jianxin Yu [2] ✉, Paul B. Wignall[1], Alexander Farnsworth [4,5], Isabel P. Montañez[6], Nian Peng[2], Qinzhong Liang[7], Xin Sun[2], Benjamin J. W. Mills [1] & Barry H. Lomax [8]

The Permian–Triassic mass extinction represents the most severe loss of biodiversity in Earth history and profoundly reorganized terrestrial ecosystems. On land, this crisis was followed by a marked floral turnover, with herbaceous lycophytes dominating Early Triassic vegetation. Here we show that these pioneer (so-called disaster) taxa that rapidly colonized stressed post-extinction environments, possessed specialized physiological traits that promoted survival under extreme conditions. Independent phylogenetic analyses show that Early Triassic lycophytes are closely related to modern Isoetales, a group characterized by exceptional ecophysiological flexibility. Their carbon isotope signatures resemble those of extant *Isoetes* that use crassulacean acid metabolism (CAM) photosynthesis, indicating a similar physiological strategy in deep time. Coupling these results with climate simulations suggests that CAM photosynthesis could have conferred a substantial advantage under Early Triassic super greenhouse conditions. Together, our findings identify CAM physiology as a potential mechanism enabling plant survival and ecosystem recovery following Earth's largest mass extinction.

The end of the Palaeozoic Era approximately 252 million years ago (Ma) coincides with extensive volcanism from the Siberian Traps and was marked by global climate warming and environmental changes[1–4]. This led to the Permian–Triassic mass extinction (PTME) where oceanic species extinction rates exceeded 81%, while terrestrial tetrapod genera experienced 89% losses[1]. However, the nature of terrestrial vegetation response to this major environmental change is a matter of ongoing research and contrasting perspectives[5–9]. This lack of consensus is partly due to the taphonomic influence on plant fossil preservation[9,10]. Furthermore, precise dating of terrestrial sequences is difficult, making stratigraphic correlation of floras challenging; consequently, the PTME in terrestrial records is often discussed as the Permian–Triassic transition (PTT)[5,6,10]. However, what is apparent is that the occurrence of a large-scale floral turnover at the PTT was followed by a distinct, low diversity and low abundance lycophyte-dominated community (Fig. 1)[5,6,11,12]. Across a broad span of latitudes, from equatorial South

[1]School of Earth and Environment, University of Leeds, Leeds, UK. [2]State Key Laboratory of Geomicrobiology and Environmental Changes, School of Earth Sciences, China University of Geosciences, Wuhan, People's Republic of China. [3]School of Geography, Earth and Environmental Sciences, and Birmingham Institute of Forest Research, University of Birmingham, Birmingham, UK. [4]School of Geographical Sciences, Cabot Institute for the Environment, University of Bristol, Bristol, UK. [5]State Key Laboratory of Tibetan Plateau Earth System, Environment and Resources (TPESER), Institute of Tibetan Plateau Research, Chinese Academy of Sciences, Beijing, People's Republic of China. [6]Department of Earth and Planetary Science, University of California, Davis, CA, USA. [7]School of Computer Science, China University of Geosciences, Wuhan, People's Republic of China. [8]School of Biosciences, University of Nottingham, Loughborough, UK. ✉e-mail: z.xu@leeds.ac.uk; yujianxin@cug.edu.cn

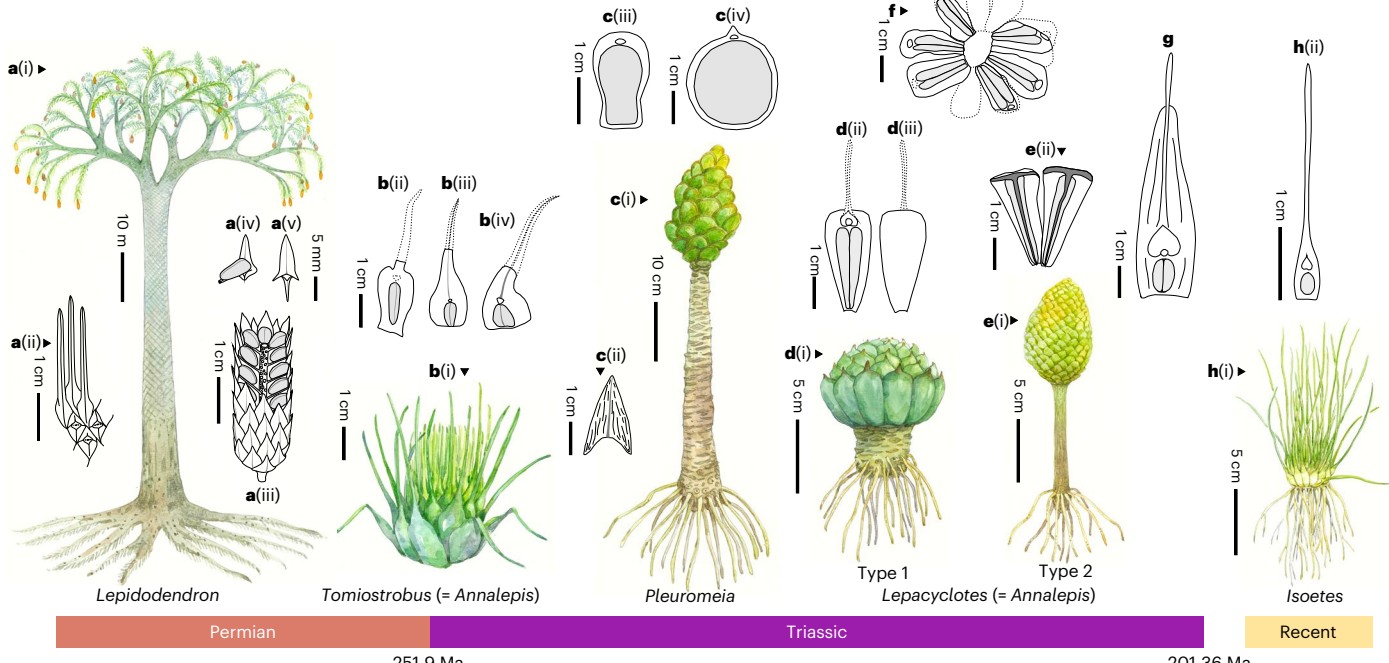

**Fig. 1 | Representative lycophytes reconstructions from Late Permian to recent. a**, (i), *Lepidodendron* reconstruction; (ii), leaf and leaf scar of *Lepidodendron*; (iii), *Lepidodendron* strobile and sporangia; (iv), *Lepidodendron* sporophyll and sporangia; (v), *Lepidodendron* sporophyll. **b**, (i), *Tomiostrobus* reconstruction, modified after ref. 13; (ii) to (iv), *Tomiostrobus* (= *Annalepis*) sporophyll with sporangia from Permian–Triassic transitional Kayitou Formation in South China. **c**, (i), *Pleuromeia* reconstruction based on in situ *Pleuromeia* fossil from Middle Triassic Badong Formation in South China; (ii), *Pleuromeia* vegetative leaf from Middle Triassic Badong Formation in South China; (iii), *Pleuromeia sanxiaensis* sporophyll with sporangia from Middle Triassic Badong Formation in South China; (iv), *Pleuromeia marginulata* sporophyll with sporangia from Middle Triassic Badong Formation in South China. **d**, (i), One possible reconstruction of the *Lepacyclotes* (= *Annalepis*) based on in situ fossils from Middle Triassic Badong Formation in South China; (ii), adaxial side of the *Lepacyclotes* (= *Annalepis*) sporophyll with sporangia; (iii), abaxial side of the *Lepacyclotes* (= *Annalepis*) sporophyll. **e**, (i), Another possible reconstruction of the *Lepacyclotes* (= *Annalepis*) based on in situ fossils from Middle Triassic Badong Formation in South China; (ii), adaxial side of the *Lepacyclotes* (= *Annalepis*) *zelleri* sporophyll with sporangia from Middle Triassic Badong Formation in South China. **f**, *Lepacyclotes* (= *Annalepis*) *zelleri* sporophyll assemblage in circle from Middle Triassic Badong Formation in South China. **g**, adaxial side of the *Lepacyclotes* (= *Annalepis*) *brevicystis* sporophyll with sporangia from Middle Triassic Badong Formation in South China, modified after ref. 29. **h**, (i), *Isoetes* sketch; (ii), adaxial side of the *Isoetes* sporophyll with sporangia. The grey circle inside the sporophyll shows sporangium.

China to high-latitude Siberia, the rise to dominance of the herbaceous lycophyte *Tomiostrobus* coincided with the extinction of the previously dominant Palaeozoic taxa, including *Gigantopteris* and *Cordaites* during the PTT (Fig. 1)[5,6,12–15].

For approximately 5 million years (Myr) after the PTME, the Earth experienced extreme warmth, with equatorial sea surface temperature over 35°C and equatorial land surface temperatures over 45°C (ref. 3,16), linked to at least a fourfold increase in atmospheric $CO_2$ concentration to over 2,600 p.p.m. (refs. 4,17,18). These conditions exceed both the photosynthesis optimal temperature threshold and $pCO_2$ saturation point for modern $C_3$ plants[19–22]. The near-total dominance of herbaceous lycophytes in lowland settings in the post-extinction interval implies that they were perhaps uniquely adapted to these extreme earliest Mesozoic climates and environments[8,11,12,23,24]. Understanding the specific traits that conferred survival advantages to these lycophytes is of critical importance to unravel the elusive killing mechanisms—and might provide insights for predicting future biosphere evolutionary trends under severe warming scenarios.

Our current understanding of these pioneer herbaceous lycophytes from the PTT is limited due to inconsistency in their taxonomy[14,15,25–28]. These plants are structurally simple, and their stems, leaves and roots are typically indistinguishable from one another, but fortunately their sporophylls (fertile, sporangium-bearing leaves) are character rich (but also morphologically variable) thereby allowing distinct species and genera to be distinguished[13–15,28]. But this combination of factors makes the identification of taxa from individual

plant specimens problematic, leading to poorly resolved taxonomy and a limited understanding of their phylogeny. As an example, the widely used sporophyll genus *Annalepis* Fliche 1910 has been replaced taxonomically by *Tomiostrobus* Neuburg 1936 and *Lepacyclotes* Emmons 1856 in different studies[15,26], but whether these 'taxa' represent the same or multiple different taxa remains unresolved due to a lack of detailed analysis[25,28–30] (Supplementary Table 1). This poorly constrained taxonomy has hindered our understanding of their diversity, phylogenetic relationships, environmental importance and functionality.

Motivated by these questions, we collected data from 485 identifiable and measurable lycophyte sporophyll specimens from different regions and geological ages including living species; 285 come from late Permian to Middle Triassic strata of southwest China, and these were compared with 200 specimens recorded in the literature (Supplementary Information and Supplementary Data 1). Most of the specimens are isolated sporophylls, but for each genus there is at least one specimen representing a complete plant or a cone with sporophylls attached to the central axis (Supplementary Information). In this Article, we focus on lycophyte sporophylls because they are the most character-rich organs, show considerable phenotypic variation within and between taxa, and provide the best evidence on lycophyte diversity, phylogeny and functionality[13–15,28,31]. To quantify the morphology of individual sporophylls, we scored them for 127 binary (present/absent) morphological character states (Ch-1 to Ch-127; Supplementary Information and Supplementary Data 1) in a morphometric database;

these characters include the diagnostic features of each taxon and are also related to sporophyll function. By measuring multiple specimens of each 'taxon', we aim to characterize sporophyll heterophylly within single species and genera and plot 'taxon' morphospace using principal component analysis (PCA). Subsequently, the PCA data were used to identify a representative specimen for each 'taxon'—selected solely for analytical purposes—to serve as an anchor point in the neighbourhood network analysis (NNA) for exploring morphological similarity and phylogenetic patterns. These specimens are not proposed as formal type specimens—they are solely used for computational purposes (Supplementary Figs. 2–4) to aid reproducibility. This taxonomic analysis allows us to place the lycophyte taxa from the PTT into a broader context by comparing them to their extinct and extant relatives (see Methods for detail).

To address the potential limitation of extant relatives' traits not being inherited from their extinct ancestors, morphological/morphometric data are supplemented by carbon isotope data to infer variation in the photosynthetic pathway[24,32] of lycophytes from the PTT taxa. Carbon isotope data have been collected from individual sporophylls and the sediment surrounding them to ensure that we have sampled the fossil itself, rather than recovering a signal of dispersed carbon from the host sedimentary rock (Supplementary Fig. 6).

Then the latest version of the coupled Hadley Centre Earth System Model version 3 with a low-resolution ocean performed by the BRIDGE group (HadCM3BL) climate model is used to simulate both average and maximum daily land surface temperatures[3]. By integrating these climate simulations with the spatial and temporal distribution of fossil occurrences, we evaluate the physiological viability of these lycophytes under extreme greenhouse conditions.

## Morphological phylogeny of herbaceous lycopods

Sporophyll morphological variability was encoded in the numerical character matrix (Supplementary Information and Supplementary Data 1) and visualized through two-dimensional PCA. Polygon areas within the PCA were used to determine the heterophylly of each taxon, resulted from the development stage or level of maturity growing position on the plant, or intraspecific phenotypic variation[13,25] (Fig. 2). Most direct size-related characters—such as Ch-28 to Ch-52 for sporophyll size (length, width, area) and Ch-92 to Ch-109 for sporangium size—contributed less than 0.1 to the top five principal components (character description is in the Supplementary Information, and the loading score is in Supplementary Table 3). This suggests that developmental or positional variation does not obscure taxonomic resolution within our character matrix. Visualization of the Phanerozoic lycopod sporophyll data from the Devonian to recent (Fig. 2b) reveals that Mesozoic taxa occupied a distinctly different morphospace to their Palaeozoic relatives (Fig. 2b)[14,33].

Within the Mesozoic lycopods, the herbaceous genera *Annalepis*, *Tomisotrobus* and *Lepacyclotes* were most common both spatially and temporally[11,14,15,26,28], especially in South China[5,6,29]. The PCA analysis was initially focused on taxa from South China to avoid the possibility of convergent evolution of taxa from different regions occupying similar climate space. Following this, the analysis was conducted on the global dataset to test for a palaeo-phytogeography signal.

The PCA of lycophyte flora during the PTT to the Middle Triassic in South China (Fig. 2a) reveals important morphological overlaps and distinctions. Specifically, PCA of the Permian–Triassic transitional *Annalepis* share a substantial overlap in morphospace with *Tomiostrobus radiatus* Neuburg, 1936 (Fig. 2a), the type species of this genus from Russia. However, a distinct boundary is observed between these taxa and the Middle Triassic species of *Annalepis*, which includes the type species *Annalepis zeilleri* Fliche, 1910 and *Lepacyclotes* Emmons, 1856. This suggests that the herbaceous lycopods of the PTT belong to the genus *Tomiostrobus* (syn. *Annalepis*) Neuburg, 1936, while the Middle Triassic lycopods are better classified under the genus

*Lepacyclotes* (syn. *Annalepis*) Emmons, 1856. This is further supported by global analyses of sporophyll clusters, as shown in Fig. 2b (ref. 5,15,25,26).

Our PCA also reveals that Permian–Triassic transitional lycophytes from South China, within the *Tomiostrobus* (*Annalepis*) group, occupy two distinct morphospaces (Fig. 2a). *Tomiostrobus brevicystis* is confined to the right side (upper and lower quadrants), while the left side (upper and lower quadrants) includes *Tomiostrobus zeilleri*, *Tomiostrobus angusta* and two unidentified species (Fig. 2c). The observed overlap in PCA space suggests that *T. zeilleri*, *T. angusta* and the unidentified taxa may represent a single taxon (*T. zeilleri* comb. nov.), with *T. brevicystis* clearly distinct. Global distribution PCA of *Tomiostrobus* sporophylls identifies four clusters (Fig. 2d). Three clusters are situated along principal coordinate 2, with two low-latitude clusters—*T. zeilleri* comb. and *T. brevicystis*—and mid-to-high latitude clusters from Xinjiang (north-west China) and Russia. A fourth cluster, on the left, represents high-latitude taxa from Russia, Greenland and Australia. The overlapping morphospace of high-latitude taxa from the northern and southern hemispheres indicates possible convergent evolution driven by similar climatic conditions, and/or they are polar remnants of a previously widespread ancestor.

PCA of Middle Triassic *Lepacyclotes* from South China (Fig. 2e) reveals three distinct clusters: one combining *Lepacyclotes angusta* and *Lepacyclotes latiloba* within the morphospace of *Lepacyclotes zeilleri* (proposed as *L. zeilleri* comb.), a second cluster representing *Lepacyclotes brevicystis* and a third cluster for an undescribed *Lepacyclotes* sp. 2 with minor overlap with *Lepacyclotes zeilleri* comb. PCA of global *Lepacyclotes* occurrences (Fig. 2f) reveals six broad groupings that partially overlap one another but lack the clear separation between groups as seen in the Permian–Triassic transitional *Tomiostrobus*, indicating greater diversification of *Tomiostrobus*. The diversification of *Tomiostrobus* might indicate that the genus existed before the PTME when climatic conditions were more varied[11]. Presumably they grew in isolated communities within stressed environments with poor preservation potential, for example, in and around mountain lakes[34]. Or alternatively, it could indicate the rapid radiation of *Tomiostrobus* in the early stage of warming. By contrast, the similarity in morphology of later-evolved *Lepacyclotes* across different geological basins and latitudes might reflect the globally weakened latitudinal temperature gradients under the hothouse conditions following the PTME.

Our data reveal clear clustering of taxa in PCA space and highlight a degree of morphological variation within each taxon. To further explore the phylogenetic context for these observations, the specimen closest to the centroid of each PCA polygon (Supplementary Information) was taken as the most representative example of that taxon, and data from this individual was used to perform NNA; these taxa represent 'voucher' samples and are identified and illustrated in Supplementary Figs. 2–4. The NNA results are in line with data from our PCA identifying 12 distinct genera of lycopod sporophyll throughout the Phanerozoic. Within the Isoetales, *Tomiostrobus*, *Lepacyclotes*, *Isoetities* and *Isoetes* all belong to the family Isoetaceae, while *Pleuromeia*, *Cyclostrobus* and *Lycostrobus* belong to the family Pleuromeiaceae; the Permian–Triassic genus *Tomiostrobus* has the closest phylogenic similarity with recent *Isoetes* (Fig. 3).

## Carbon isotopes of latest Permian to Middle Triassic herbaceous lycopods

The carbon isotope composition ($\delta^{13}C$) of extant and extinct plants has been successfully used to identify photosynthetic pathways and environmental stresses[24,32,35,36]. However, carbon isotope fractionation within the same plant species can vary under different climatic and environmental conditions, particularly due to differences in water availability[37]. Accordingly, we restricted our carbon-isotope analyses to latest Permian–Middle Triassic lycophytes preserved in

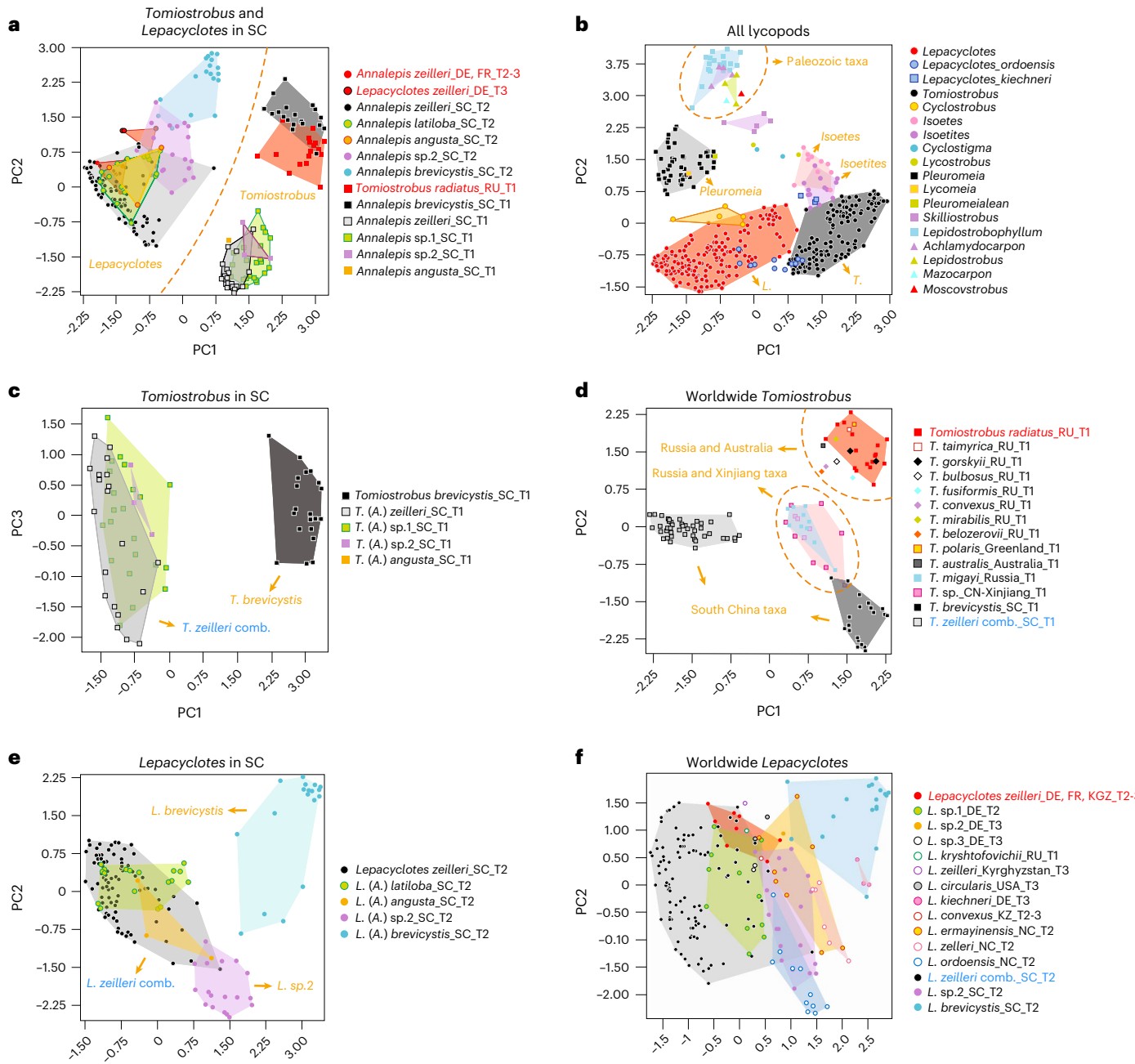

**Fig. 2 | Two-dimension PCA result of all the lycopods sporophyll morphology.** **a**, *Tomiostrobus* and *Lepacyclotes* sporophyll in South China; together with the type species *Tomiostrobus* (= *Annalepis*) *radiatus* in Russia and *Lepacyclotes* (= *Annalepis*) *zeilleri* in Germany and France. **b**, Representative lycopod sporophyll from the whole Phanerozoic. **c**, *Tomiostrobus* sporophyll in South China. **d**, Worldwide *Tomiostrobus* sporophyll. **e**, *Lepacyclotes* sporophyll in South China. **f**, Worldwide *Lepacyclotes* sporophyll. The name of each sporophyll group in the legends of figures **a**–**e** shows the genera name (old genera name) _species name_the international abbreviation of the depositing area_age.

Sporophyll names in red indicate the type species of each genus. L., *Lepacyclotes*; T., *Tomiostrobus*; A., *Annalepis*; DE, Germany; FR, France; SC, South China; RU, Russia; KGZ, Kyrgyzstan; KZ, Kazakhstan; NC, North China. Higher-resolution vector figures can be reproduced from the morphometric data in Supplementary Data 1 using either the PCA code in the Supplementary Information or the free software PAST (https://palaeo-electronica.org/2001_1/past/pastprog/index. html). Note that, for each subfigure, the corresponding data in Supplementary Data 1 must be appropriately filtered.

coastal lowland deposits of South China (Supplementary Fig. 6). During the study interval, South China was in a low-latitude tropical region with limited temperature seasonality[6], thus minimizing the influence of climatic factors such as temperature fluctuations and water-use efficiency on plant physiology and associated carbon isotope signatures. By plotting the sporophyll morphometrics (Supplementary Fig. 7) and $\delta^{13}$C values (Supplementary Fig. 8) against sedimentary facies for end-Permian to Middle Triassic South China plants, we find

that neither taphonomy nor growth environment (for example, differences in salinity) is the primary control on carbon-isotope fractionation or taxonomic assignment. Rather, geologic age (reflecting background atmospheric $CO_2$) and genus exert stronger influences.

In modern trees, intra-organ variation in carbon isotope values— for example, from the mid-vein to the leaf margin—can reach up to 3 (ref. 38). To reduce such internal variability in fossil samples, we collected material from the entire organ whenever possible. This approach

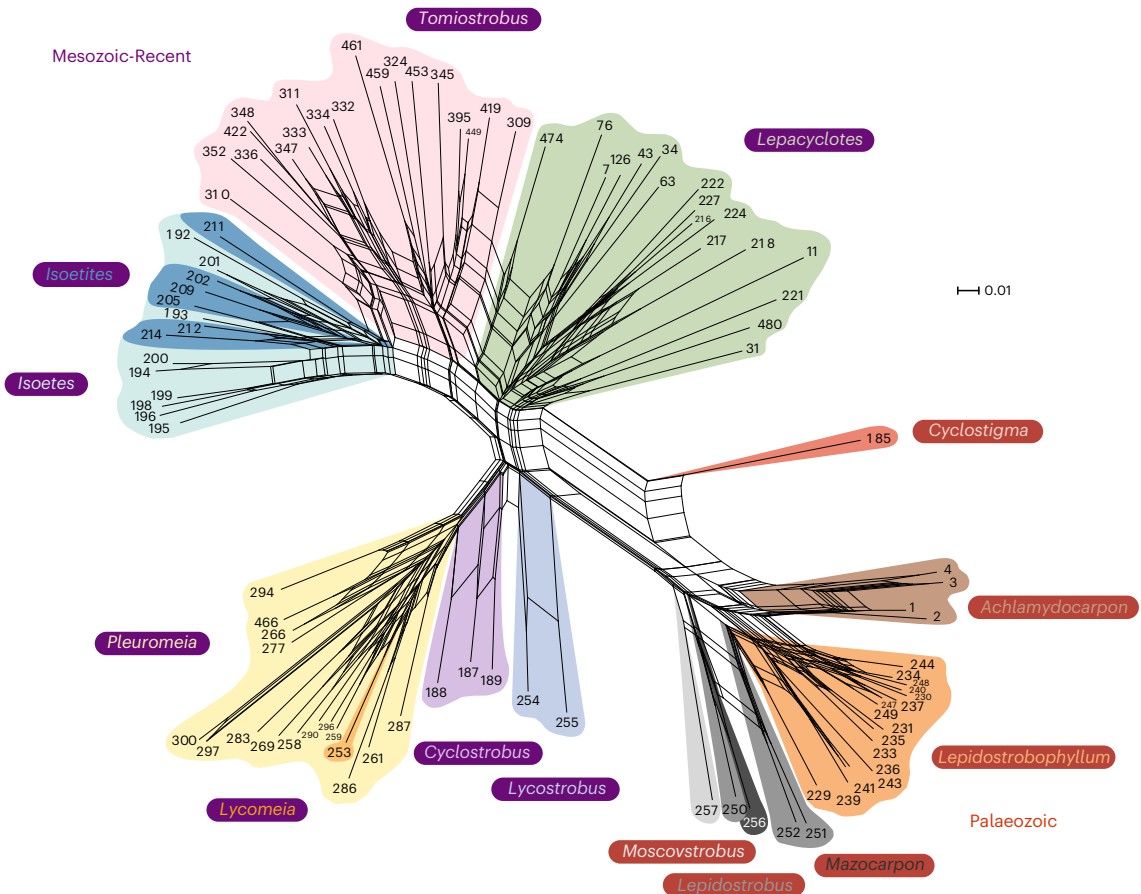

**Fig. 3 | Neighbour-net of all lycopods species sporophyll.** Each number at the end of each branch represents a lycopod sporophyll species. The data of each species come from the best-preserved sample among all the specimens. The number-species comparison could be seen in the Supplementary Information.

*Isoetes*, the recent species known for the facultative CAM photosynthesis; *Tomiostrobus*, Permian–Triassic transitional taxa. See Methods for reproducing the higher-resolution vector figure.

is essential because Permian–Triassic plants are generally small, and even a single fossil specimen often does not yield sufficient organic material for $\delta^{13}C$ analysis (see specimen pictures and scale in the Supplementary Information). Furthermore, to avoid contamination from host sediment, only the exposed surface of each fossil was sampled. For extremely small or thin-cuticle plants—such as the PTT seed fern *Germaropteris* and Triassic lycophytes—specimens of the same species, from the same locality and stratigraphic layer, were pooled to obtain sufficient carbon for analysis. This resulted in fewer but higher-quality data points (Fig. 4).

Similar $\delta^{13}C$ values of organic matter in the matrix associated with each plant fossil in near-shore sedimentary facies from different locations suggests that the plant fossils are likely penecontemporaneous (see Supplementary Figs. 6a and 6d for details). A few literature-derived plant $\delta^{13}C$ values lacking corresponding sedimentary background data were considered unreliable and therefore excluded from our main analysis, though they are listed in the Supplementary Information for reference and comparison.

The result shows the late Permian pre-extinction arborescent lycopod *Lepidodendron*, the conifer *Anshuncladus* and other plants all share a similar $\delta^{13}C$ value of about −24.6‰ (Fig. 4), indicating a similar physiology in carbon isotope fractionation, likely C$_3$ (refs. 39,40). After extinction, the mean $\delta^{13}C$ values of the non-lycophyte flora are more negative (approximately −30.5 ± 1.0‰), tracking the isotopic shift in global atmospheric $\delta^{13}C$ ($CO_2$)[1,2,4,16]. By contrast, the mean $\delta^{13}C$ composition of the Permian–Triassic transitional lycophyte *Tomiostrobus* flora is ~3.4 ± 0.61‰ (~1.2‰ to ~6.5‰) higher than the contemporaneous non-lycophyte flora (Fig. 4). The extreme environ-

mental conditions after the PTME led to a 5 Myr coal gap and scarcity of Early Triassic terrestrial plant fossils[41], leading to a relatively sparse dataset for this part of the study. Therefore, we were unable to conduct carbon isotope comparisons at the family level. Rather, we performed broader comparisons between lycophyte and non-lycophyte taxa, which may introduce potential broader uncertainties linked to differences in phylogeny, growth environments and post-depositional processes. The Middle Triassic *Lepacyclotes-Pleuromeia* lycophyte flora have median $\delta^{13}C$ values 0.73 ± 0.41‰ higher than contemporaneous non-lycophytes (including *Neocalamites*, *Voltzia*, megaphyllous leaf with *Spirorbis*, indeterminate conifer and indeterminate seeds) (Fig. 4 and Supplementary Fig. 6). Considering the ~4-fold increase in $pCO_2$ and substantial temperature increase in the Early Triassic[4,16], the relative carbon isotope stability of these herbaceous lycophytes is remarkable. The fossil material selected for carbon isotope analysis was confirmed to be well-preserved cuticle, based on fluorescence microscopy observations that revealed epidermis-like cellular structures (Supplementary Fig. 34), supporting the interpretation that the $\delta^{13}C$ values reflect original plant tissue rather than recalcitrant diagenetic residues.

Nevertheless, the multiple factors influencing carbon isotope fractionation—particularly the large natural variability in the isotopic composition of source materials including atmospheric $CO_2$, $CO_2$ derived from sediment organic matter decomposition and dissolved inorganic carbon in water—introduce considerable uncertainty, thereby limiting the extent to which our isotope data can be used to calculate crassulacean acid metabolism (CAM) productivity directly. To place our morphological and isotopic results to a broader context,

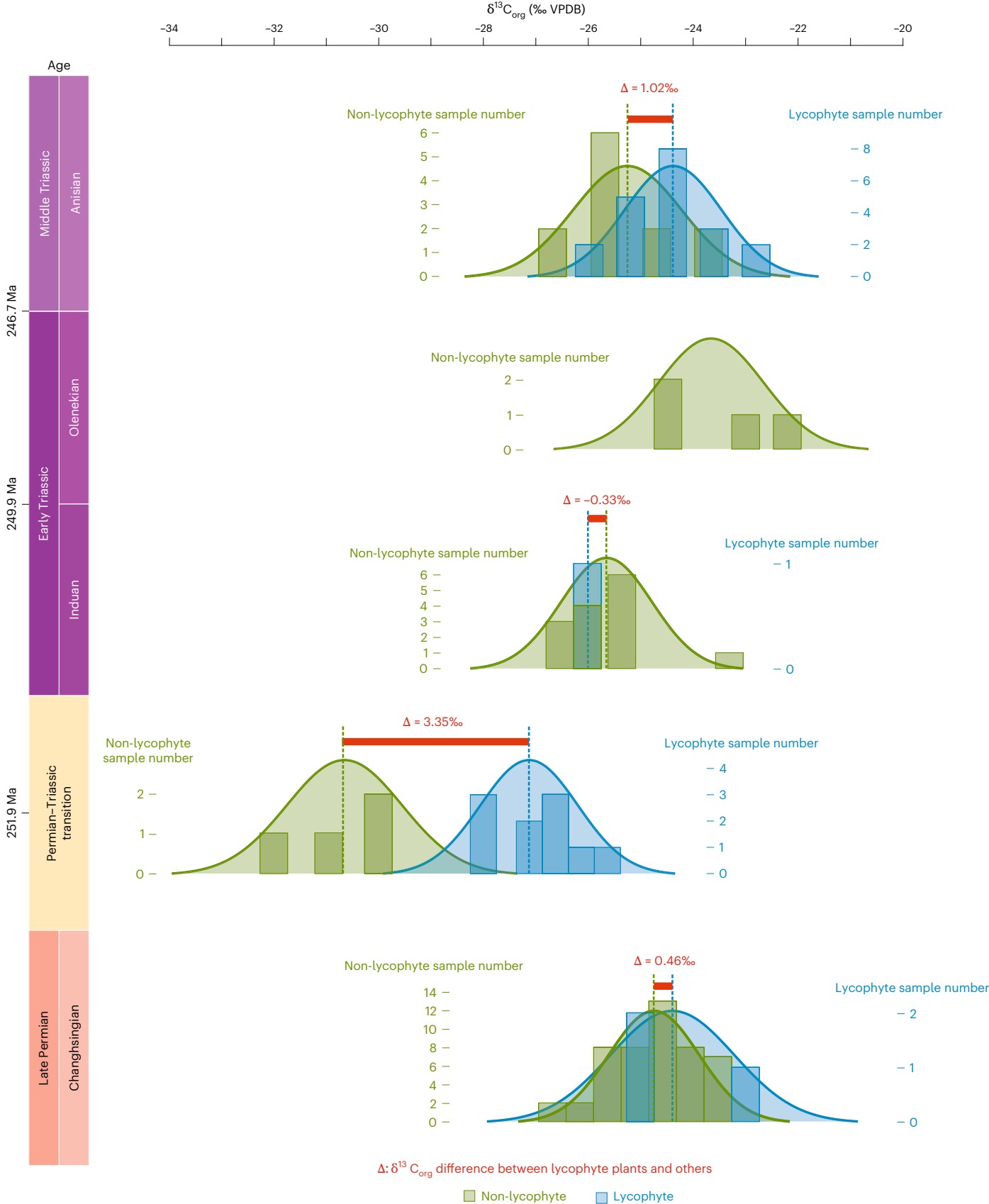

**Fig. 4 | Organic carbon isotope values of the plant fossil and carbon isotope difference between the lycophyte plants and other plants from end Permian to Middle Triassic in South China.** See details in Supplementary Fig. 6, the original data in Supplementary Table 5 and the sampling pictures of the fossils in Supplementary Figs. 24–33. VPDB, Vienna Peedee Belemnite.

we used the Earth system model HadCM3BL to simulate palaeo-climate conditions—particularly changes in land surface temperature—across the PTT[3]. By coupling these simulations with the known fossil occurrences of Triassic lycophytes, we aim to more broadly evaluate whether extreme thermal conditions could have necessitated the use of CAM photosynthesis for survival.

## HadCM3BL climate simulation

The Earth system model HadCM3BL is capable of simulating robust climate conditions for the Permian–Triassic interval, consistent with multiple climatic and environmental proxy records[3]. Using this model, we generated maps of both average and absolute maximum daily land surface temperatures for three key intervals: the end-Permian Changhsingian (pre-PTME; Fig. 5g,h), the PTT (syn-PTME; Figs. 5d,e) and the Early Triassic Induan (post-PTME; Fig. 5a,b), under reconstructed atmospheric $CO_2$ concentrations, sea surface temperature proxies and climatic facies and mineralogical data (see detailed explanation in 'HadCM3BL climate simulation' in Methods).

By overlaying palaeogeographically corrected macrofossil and microfossil records of Triassic lycophytes onto these palaeogeographic maps (Fig. 5c,f,i), we determined the modelled average and maximum land surface temperatures at each fossil locality. Fossil evidence shows that lycophytes were most widespread during the PTME, with many occurrences located between 45° N and 80° S where average maximum daily land surface temperatures exceeded 40 °C (Fig. 5).

Extant $C_3$ plants have an optimal growing temperature of 10–35 °C and are unable to survive at higher temperatures due to physiological constraints such as water limitation, Rubisco enzyme deactivation and elevated photorespiration[19,22,42,43]. By contrast, these Triassic lycophytes were able to persist in regions such as South China, North China, Xinjiang, Europe, Australia, India and Argentina, where the modelled average maximum daily temperature exceeded 40 °C and the absolute maximum daily temperatures ranged from 45 °C to 65 °C (Fig. 5). One potential photosynthetic pathway that could accommodate such high daily temperatures is $C_4$ given that plants using this pathway are known for their drought and heat tolerance[19,44]. $C_4$ plants, however, are restricted to the angiosperm clade, with the earliest records dating to the Oligocene, and did not exist during the Permian–Triassic[44]. Alternatively, CAM photosynthesis has been previously hypothesized in deeper time[11,32,36,45–47]. CAM plants—dominant in recent hot, semi-arid to arid regions worldwide, including deserts—can persist under conditions with surface temperatures approaching 70 °C (refs. 21,22,48). The survival of Triassic lycophytes under comparable extreme heat is therefore more consistent with the possibility of an alternative photosynthetic pathway, such as CAM, rather than solely the $C_3$ type.

## Discussion

The PCA and NNA results indicate a close morphological linkage and thus a close phylogenic relationship between the Permian–Triassic transitional lycophytes and recent *Isoetes*. This is especially clear when comparisons are made to *Tomiostrobus*, a stratigraphically confined Permian–Triassic transitional species (Figs. 2 and 3). The morphological similarity between the two taxa is driven by a number of key structures shared between these temporally disparate sporophylls, including a herbaceous growth form (Ch-3), sporophylls arranged in compact clusters along the cone axis (Ch-11), a long and slender leaf apex (Ch-58, Ch-59), a wide side angle (Ch-73), a hastate (spearhead-shaped) leaf base that indicates a tight attachment to the central axis (Ch-80), the presence of a prominent longitudinal vein in the leaf apex (Ch-82), a clavate (club-shaped) sporangium (Ch-88) and relatively small sporangium size (Ch-99) (see detailed character explanation in the Supplementary Information with annotation figures and the biplot in Supplementary Information)[14,15]. In extant *Isoetes*, these features are associated with increased buoyancy facilitating

sporophyll transportation and dispersal through water[14,29]. It is likely that *Tomiostrobus* had the same traits as *Isoetes* which permitted its widespread spatial distribution along continental margins (Fig. 5c,f)[14,15,28–30].

The close phylogenetic relationship between extant *Isoetes* and the Permian–Triassic transitional lycophyte flora allows us to hypothesize about the factors that favoured their proliferation during the PTT. Extant *Isoetes* are mostly semi-aquatic to aquatic and are renowned for ecophysiological flexibility regarding their photosynthetic pathway (facultative CAM) and their capacity to absorb $CO_2$ from sediments and the water through their roots (passive diffusion)[46,49–51]. The pre-extinction arborescent lycophytes, such as *Lepidodendron*, similar to extant *Isoetes* of the same class, had abundant aerenchyma tissues inside their trunk which in extant *Isoetes* enables the transportation and storage of $CO_2$ as malic acid for allowing CAM photosynthesis, indicating the potential of CAM within the class Lycopiosida[23]. CAM has also been inferred in the Late Triassic *Mesenteriophyllum*, a Pleuromeiacea from polar regions that lacks stomata and thus has been assumed to have relied on $CO_2$ absorbed through its roots and CAM photosynthesis[11]. Together, these morphological, physiological and ecological parallels indicate strong evolutionary conservatism within the lycophyte clade[33], supporting the hypothesis that CAM capability could have persisted over geological timescales[46].

In extant facultative CAM species, such as *Isoetes*, the proportion of photosynthate derived from the CAM pathway increases with stress, whereas under low-stress conditions they function primarily as $C_3$ plants[36,46]. During CAM, stomata remain closed during the hot period of the day to reduce water loss during respiration and photosynthesis[48]. At night, when temperatures drop, they open their stomata to absorb $CO_2$, storing it as malic acid in vacuoles, which is used for photosynthesis during the day[49–52]. This adaptation helps these facultative CAM plants survive in hot, arid conditions and reduces photorespiration by concentrating $CO_2$ (refs. 21,22,45,48,53).

Theoretically, switching between photosynthetic pathways impacts the carbon isotopic signature of plant tissues, with facultative plants from more equable environments having a typical $C_3$ isotopic signature and stressed plants having a less negative (more enriched) $\delta^{13}C$ value due to rising CAM contribution[45]. However, *Isoetes* absorbs a portion of its $CO_2$ from sediment-derived sources with typically more negative $\delta^{13}C$ values, which can offset the expected positive shift in *Isoetes* $\delta^{13}C$ due to CAM[34,45]. As a result, *Isoetes* tends to show $\delta^{13}C$ values comparable to those of $C_3$ plants[24,53,54]. For example, the extant aquatic species *Isoetes howellii*, found in standing lakes, has a $\delta^{13}C_{org}$ of -29‰ (±0.9‰)[53]. Conversely, *Isoetes* from more water-stressed environments such as the seasonally drought-tolerant *Isoetes* (*Stylites*) *andicola* has a $\delta^{13}C_{org}$ value of −22.5‰ (ref. 55), potentially reflecting a higher proportion of photosynthesis via stress-induced CAM, although the nature of this stress fractionation response is yet to be fully characterized[24,34,45,46,53,55].

Atmospheric $CO_2$ concentration potentially increased from fourfold to sixfold during the PTT[4], accompanied by a notable negative C isotope excursion. In South China, there is a -6.5‰ negative shift in the bulk organic $\delta^{13}C$ values[16]. Similar trends are observed globally, including a general 4‰ to 8‰ negative shift in total organic carbon $\delta^{13}C$ values in terrestrial sediments and plant tissues (down to −32‰), and a -3.5‰ decrease in marine carbonate $\delta^{13}C$ values (to −1‰)[1,16,18]. During this transition, the lycophyte $\delta^{13}C_{org}$ values from this study (lycophyte $\delta^{13}C_{org}$ −27.2 ± 1.2‰) are notably less negative than those of non-lycophyte vegetation (non-lycophyte $\delta^{13}C_{org}$ −30.5 ± 1.0‰) and are closer to the $\delta^{13}C_{org}$ values of associated sediments (−27.6 ± 1.3‰) (Supplementary Fig. 6). The pronounced negative shift in $\delta^{13}C_{org}$ values of non-lycophyte plants and associated sediments is consistent with previous records, reflecting a major disturbance in the global carbon cycle. The consistent -1‰ difference between each Triassic lycophyte specimen (black dots, Supplementary Fig. 6) and the surrounding matrix (red dots, Supplementary Fig. 6) confirms that these values

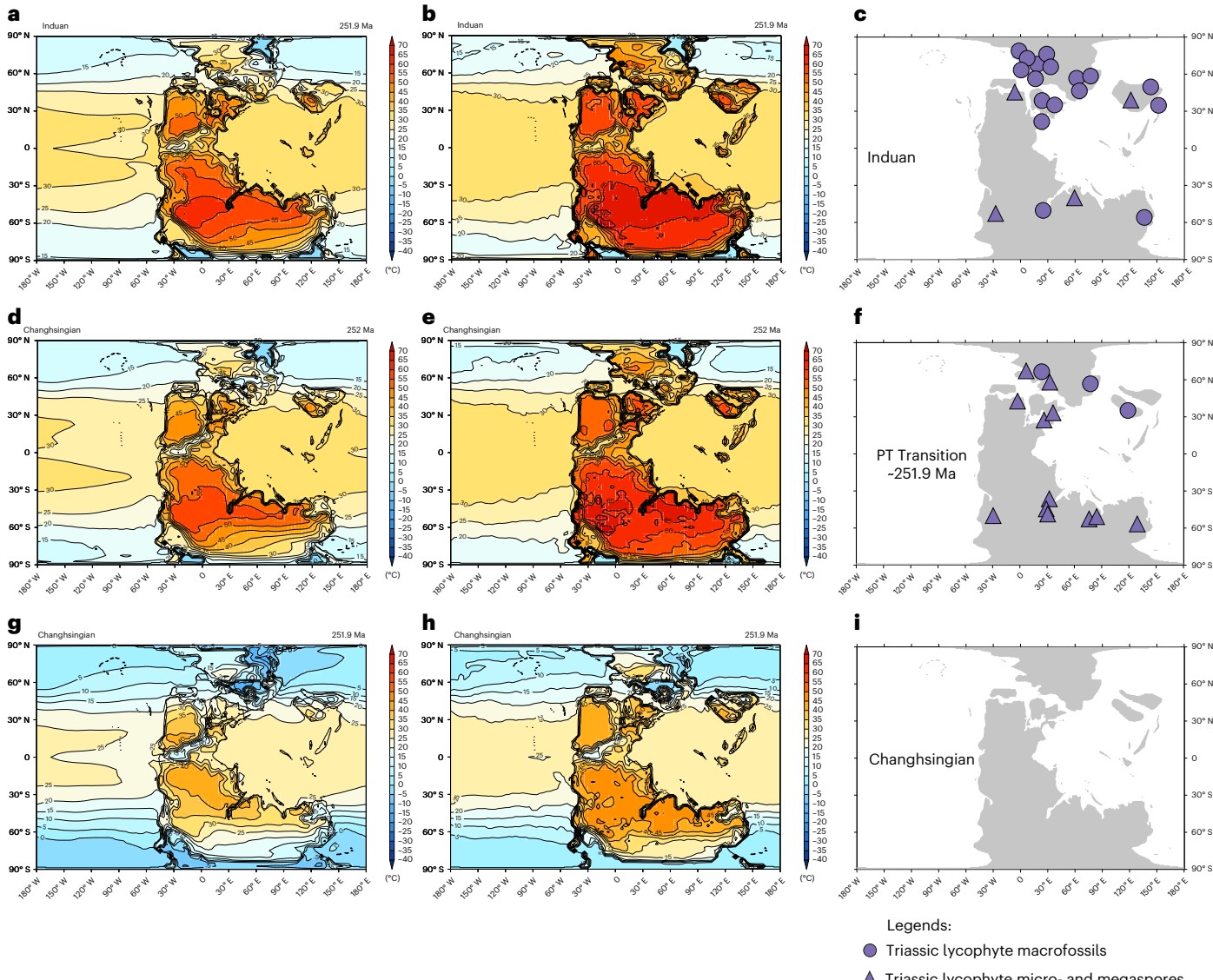

**Fig. 5 | Macrofossil and microfossil occurrences of Triassic lycophytes and HadCM3L-simulated land surface temperatures. a**, Induan average maximum daily land surface temperature (4,000 p.p.m. $CO_2$). **b**, Induan absolute maximum daily land surface temperature (4,000 p.p.m. $CO_2$). **c**, Induan (Early Triassic) distribution of lycophyte macrofossil and microfossils. **d**, Transitional average maximum daily land surface temperature (2,568 p.p.m. $CO_2$). **e**, Transitional absolute maximum daily land surface temperature (2,568 p.p.m. $CO_2$). **f**, Permian–Triassic transitional distribution of lycophyte macrofossils and microfossils. **g**, Changhsingian average maximum daily land surface temperature (412 p.p.m. $CO_2$). **h**, Changhsingian absolute maximum daily land surface temperature (412 p.p.m. $CO_2$). **i**, Changhsingian (end Permian). In grid cell with both micro and macro lycophyte fossils, we only plot the macrofossils. The macrofossil records come from this study, and the microfossils data are from refs. 11,75 and references therein.

represent primary plant material rather than diagenetic alteration (Supplementary Fig. 6).

Extant *Isoetes*, using the facultative CAM pathway, partially use sediment-derived $CO_2$, which is typically $^{13}C$-depleted compared to the air, as a substrate for carbon assimilation[34,45,53]. The more negative $\delta^{13}C$ of sediment $CO_2$ offsets the $^{13}C$-enrichment associated with the CAM pathway, resulting in a $\delta^{13}C$ composition of *Isoetes* that can overlap with those of $C_3$ plants relying on atmospheric $CO_2$ for carbon assimilation[34,45,53]. If the Triassic lycophytes used purely the $C_3$ photosynthetic pathway and assimilated only sediment-derived $CO_2$, then their $\delta^{13}C$ values would be more negative than that of the sediments. Conversely, if they used CAM photosynthesis with exclusively atmospheric $CO_2$, their $\delta^{13}C$ values would be expected to exceed those of both contemporaneous non-lycophytes (including the Permian–Triassic transitional *Germaropteris* leaf, Middle Triassic *Neocalamites*, *Voltzia*, megaphyllous leaf with *Spirorbis*, indeterminate

conifer and indeterminate seeds) and the surrounding sediments. Therefore, the observed $\delta^{13}C_{org}$ values of the Triassic lycophytes—relatively enriched compared to non-lycophytes, but similar to those of associated sediments—suggests a distinct carbon isotope fractionation pattern associated with CAM photosynthesis involving partial uptake of sediment $CO_2$ or a higher proportion of $C_3$ relative to CAM photosynthesis (see Supplementary Fig. 9 for detailed analysis).

Although certain identification of present-day CAM photosynthesis in plants is linked to nighttime malic acid accumulation, this cannot currently be tested for in fossil plants. Our carbon isotope data, however, when combined with our phylogenetic analysis and climate modelling, is most parsimoniously interpreted as evidence of Permian–Triassic transitional lycophytes using CAM as an adaptive mechanism to cope with harsh earliest Triassic climate[1,3,4,16,56]. The Permian–Triassic transitional herbaceous lycophytes that dominated coastal habitats have elevated $\delta^{13}C$ compositions relative to

contemporaneous non-lycophyte plants (Fig. 4). The difference in carbon isotope compositions between the contemporaneous floras (lycophyte compared to non-lycophyte) is at its greatest in the *Tomiostrobus* Permian–Triassic transitional flora and declines through the earliest Triassic (Fig. 4). We suggest this isotopic shift records a gradual transition to a less stressful climate[3] and a reduction in the utilization of CAM by plants which can operate both $C_3$ and CAM photosynthesis facultatively. However, the scarcity and poor preservation of plants through this time interval results in a very limited fossil record, so this assertion cannot be fully tested at present.

Extant *Isoetes* provide insights into how post-PTME lycophytes such as *Tomiostrobus* may have thrived. Some species (for example, *Isoetes piedmontana*) switch between $C_3$ and CAM photosynthesis depending on seasonal stress intensity and retreat to a corm under extreme drought or heat exceeding their highest tolerance[34,50,51]. Others (for example, *Isoetes sinensis*) use antioxidant enzyme systems to withstand desiccation and heavy-metal stress[51,52,57-61]. Some Triassic lycophytes, such as *Tomiostrobus* from South China, are interpreted as inhabiting paralic settings, akin to modern tidal-shore relatives (for example, *Isoetes riparia*)[62], where emergent and submerged forms would have benefited from the thermal buffering of water. Together, these traits—including CAM flexibility[11], dormancy[51], aquatic habits[34,49,57,62] and antioxidant defenses[52,57-61]—likely contributed to the resilience of Triassic lycophytes and highlight continuity with the survival strategies of modern *Isoetes*[5,8,11,12,14,15,30].

The PTT was highly anomalous: established, geographically widespread, diverse lowland arboreal forest ecosystems[5,6,25] were rapidly replaced by low-diversity, herbaceous, lycophyte-dominated communities across the transition[5,6,8,11,12,30]. This switch marks a change in plant body size and a reduced biomass[8,49,63]. Furthermore, our phylogenetic and isotopic analyses suggest that the PTT lycophytes were able to use the facultative CAM photosynthetic pathway, and HadCM3BL climate model simulations suggest that these lycophytes managed to survive in an area with surface temperature higher than the highest tolerance of extant $C_3$ plants. A terrestrial lowland biosphere dominated by CAM plants is greatly different from one dominated by $C_3$ photosynthesis. As an example, while CAM plants have a higher $CO_2$ fixation efficiency, the storage of $CO_2$ as acids results in their relatively lower carboxylation efficiency which feeds through to lower productivity and less growth[49,50,53,64]. Even though increasing $CO_2$ after the PTME may have helped carbon assimilation efficiency of CAM plants[63], the overall productivity of these herbaceous lycophytes, resembling present-day CAM plants under chamber $CO_2$ experiments[53,63], would have been much lower than the pre-extinction ever-wet arborescent forests of the late Permian[45,49,53,65].

Consequently, the dominance of Triassic dwarf lycophytes capable of flexibly operating CAM photosynthesis would have reduced terrestrial organic carbon burial via photosynthesis and bio-weathering[8], as well as lowered nutrient fluxes to the ocean[66,67]—a feedback that would have amplified the post-PTME warming trend[68]. However, plant macrofossils alone provide only a partial view of vegetation composition across environments[5,8,10]. To capture this more realistically, vegetation models need to incorporate the CAM functional type, at least from the PTME onward, to better simulate terrestrial biomes and productivity. Such improvements are critical for robust carbon-isotope mass-balance modelling and for evaluating the broader environmental consequences within an Earth system framework.

At the same time, the persistence of CAM lycophytes can be viewed as a critical survival strategy under the extreme precipitation variability, prolonged droughts and warmth characteristic of the 'mega-El Niño' world of the Early Triassic[3]. This resilience ensured that some lowland terrestrial vegetation cover was maintained, which may have prevented an even more profound collapse of terrestrial ecosystems and a shift to extreme greenhouse conditions well beyond the ~5 Myr recovery interval[8,21,22,48,68].

## Methods

### Lycopods sporophyll character identification and measurement

The characters used to differentiate lycopods include root structure, overall plant morphology, cone (strobilus) structure, spore type and number, sporophyll characteristics and sporangium features, incorporating both terminological organ descriptions and topological measurements[31]. Our study encompasses 127 characters for Isoetales and Lepidodendrales lycopods, with a primary focus on reproductive organs—particularly sporophylls and sporangia—which are more commonly preserved in the fossil record. Although spores are widely used in lycophyte taxonomy, most are found as dispersed specimens rather than in situ, making it difficult to confidently associate them with specific plant macrofossils and resulting in substantial missing data. As the morphology of sporophylls and sporangia is already sufficient to distinguish among taxa in our dataset, we do not emphasize spore data in depth in this study with only simple classification. Future research integrating spore ultrastructures can further refine lycophyte phylogenetic relationships.

Key distinguishing characters include overall plant growth habit (Ch-3), sporophyll phylotaxy (Ch-11), presence or absence of isophylly/heterophylly (Ch-17), apex shape and presence (Ch-58, Ch-59), base shape related to sporophyll attachment (Ch-80), sporangium shape (Ch-88, Ch-89) and sporangium surface ornamentation or structure (Ch-110, Ch-111) (see the loading value of each character in Supplementary Fig. 5). Detailed explanations and figure annotations for these characters are provided in the Supplementary Information. Each specimen of every taxon is coded in a character matrix (Supplementary Data 1), with images and sketches of lycopod sporophyll fossils available in the Supplementary Information.

When selecting characters to distinguish between species, having more characters does not always improve the outcome. Speciation is influenced by isolation and adaptation to different environments. Each species comprises individuals that have evolved under similar environmental conditions, leading to the development of new morphological characters derived from ancestral traits. Therefore, selecting morphological characters with functional role is crucial, especially for studies related to plant physiology. Including too many nonfunctional characters can dilute the results and reduce their reliability. Characters inherited from common ancestors should be excluded when performing clustering within the same family or order. In addition, random characters lacking functional roles—potentially arising from genetic mutations or preservation biases rather than natural selection—should also be excluded.

In animal phylogenetics, characters are categorized and weighted based on their functional roles[69]. Similarly, in this study, we have reviewed and discussed the potential functions of the characters used to inform subsequent phylogenetic and ecological analyses. Many characters, such as sporophyll shape and sporangium position, are related to water transport capabilities, while sporophyll base shape affects the attachment and transport of sporophylls on the central axis[14,29]. Detailed functional inferences for most characters are provided in the discussion section of the Supplementary Information. However, some characters in our matrix lack clearly defined functions, a challenge exacerbated by the limited availability of close extant relatives and the recent extinction of many genera[70-72]. Given the existing gap between plant morphology and function, each character in our matrix is considered equally important[71,72].

### PCA

Two-dimensional PCA was conducted on the presence/absence of data for lycopod characters, using Euclidean distances in PAST (v4.02)[10,71]. The method effectively reveals both gradual and distinct variations in sporophyll morphology. Gradual variations are considered within-species diversity, while distinct variations are interpreted as representing different species or subspecies. To capture as much

morphological variation (heterophylly) as possible within each taxon, all available plant fossil samples were included in the PCA. In cases where fossils were incomplete but identifiable, missing portions were inferred by comparison with better-preserved specimens of the same species. Fossils that were poorly preserved with unpredictable missing parts or lacking critical information were excluded from the analysis. Consequently, some lycopods of interest may be absent from the dataset. Researchers are encouraged to follow the protocols outlined in the Supplementary Information for incorporating their own fossil collections to enhance the database.

We used original taxonomic names rather than combined or revised names to avoid conflating data and introducing potential biases. For instance, ref. 25 proposed synonymizing dispersed sporophylls previously classified as four species by ref. 30 into a single species, *Tomiostrobus sinensis*, which was excluded from our analyses.

In the PCA, each character represents an independent dimension, with data point locations determined by their Euclidean distances across these dimensions. Taxa are grouped based on all data points corresponding to a specific species or genus. For visualization, the high-dimensional taxon volumes were projected into a two-dimensional space that captures the maximum amount of character information. The summary scores for each principal component (PC), representing the percentage of variance explained, are listed in Supplementary Table 2. The top three principal components are used for generating the two-dimensional morphospace plots, with the highest score PC1–PC2 shown in Fig. 2 and additional PC1–PC3 plots in Supplementary Fig. 1. Note that only a subset of character information is included in the PCA analysis.

In the PCA plots, polygons of different colours represent clusters of sporophyll characters corresponding to individual species groups. The area of these polygons reflects the range of morphological variation within each taxon, with larger areas indicating greater variation[71]. Proximity between polygons suggests potential close relationships that warrant further NNA. Overlapping morphospaces are interpreted as potentially representing subspecies. In the PCA analyses shown in Fig. 2, certain characters present or absent in all selected taxa were excluded to prevent data dilution (highlighted as red in Supplementary Data 1). All the fossils are preserved in Room 014B, Main Building, China University of Geosciences (Wuhan).

## NNA of cladistic matrix

Neighbourhood network (NNA) is a clustering method that incorporates all characters and is widely used in phylogenetic analysis. It is particularly useful for phylogenetically unsorted taxa, such as most plant fossils, where homoplastic (incompatible) signals can overshadow phylogenetic signals, potentially leading to incorrect tree inferences[72]. Unlike dichotomous tree methods, neighbourhood network effectively handles non-tree and incompatible signals by representing them as a network, thus providing a more accurate depiction of ancestral–descendant relationships[72].

For phylogenetic NNA analyses, we selected one 'best-preserved' specimen per taxon to represent the taxon. However, given the heterophylly within taxa as illustrated by the polygon areas in our PCA results, defining the best-preserved fossil can be ambiguous. To minimize subjective bias and mitigate the influence of incompatible data, we selected only one specimen per species that was closest to the centroid of the polygons in the PCA results, reflecting the morphological variation of sporophylls within each lycopod taxon. All taxa and character matrix data were stored in Mesquite (v3.70) and uploaded into PAUP (v4) for distance matrix calculations. The resulting distance matrix was then used to generate the neighbourhood network in SplitTree (v4.18.3). For detailed procedural instructions, refer to ref. 72.

The distance between each tip in the NeighborNet represents the morphological distance between samples, with a 0.1 scale bar indicating the distance in pixels.

We compared the results of the NNA with the PCA results to ensure consistency in phylogenetic information. Both results indicate 12 independent genera of lycophyte sporophyll: Palaeozoic *Cyclostigma*, *Achlamydocarpon*, *Lepidostrobophyllum*, *Mazocarpon*, *Lepidostrobus* and *Moscovstrobus*, and Mesozoic to recent *Lycostrobus*, *Cyclostrobus*, *Pleuromeia* (*Pleuromeialean*, *Lycomeia*), *Isoetes* (*Isoetites*), *Tomiostrobus* and *Lepacyclotes* (Fig. 3). There are clear transitional taxa between each genus in the families Isoetaceae and Pleuromeiaceae. For example, *Tomiostrobus* (*Skilliostrobus*) *australis* (number 310) occurs between *Tomiostrobus* and *Isoetites*, while the *Lepacyclotes* found in North China during the Middle Triassic have the highest similarity with the Permian–Triassic transitional *Tomiostrobus angusta*. The latter is included within *Tomiostrobus zeilleri*, and *Pleuromeia shaolinii* is associated with *Pleuromeia* and *Cyclostrobus* (Fig. 3). Based on this comparison, we are able to revise the taxonomy of Triassic Isoetales lycopod sporophylls to robustly distinguish genera, species and subspecies based on our presence–absence data and our morphometric analysis (Supplementary Table 2). Our result suggests there are 26 species including 44 subspecies on a global scale, rather than the 73 species suggested by the existing taxonomy (Supplementary Table 4). Our dataset contains recent *Isoetes* species and comparable fossil lycopod species, providing a window that links fossil plants to their living descendants. This allows for an exploration of the linkage between morphology, genetics and phylogeny. In Supplementary Table 4, the red and bold taxa are distinct extant species with species designation via either morphological and/or genetic information; thus, these occurrences represent valid taxa and should not be synonymized with taxa in the same branch of the NNA tree. Comparisons between our revised taxonomic groupings based solely on morphology and the current genetically based phylogenetic species lists of *Isoetes* and *Isoetites* suggests that our dataset and data processing methods (PCA and NNA) might have artificially reduced the diversity. This is due to factors including (1) morphological characters from other parts of the plant aside from sporophylls distinguishing living species, (2) loss of morphological information during fossilization and (3) the increasing primacy of genomic information in systematics of living species. For example, in our morphological analysis the extant species *Isoetes cangae* and *Isoetes serracarajensis* resolve as a single species, whereas molecular analysis identifies them as distinct species[73]. Overall, these results suggest that our morphology-based phylogeny is, predictably, of lower resolution than a genetic-based taxonomic system, especially in closely related species with similar sporophyll organization. However, this integration of extant and extinct plants into a single phylogenetic framework allows us to pose new questions about the ecophysiology of these extant floras.

## Carbon isotopes

Carbon isotope ratios reflect the balance of physiological processes in plants, such as photosynthesis, respiration and transpiration over the lifetime of that tissue. These processes are influenced by atmospheric $CO_2$ pressure, temperature, and local environmental factors such as water availability and salinity. To accurately separate physiological differences in palaeo-plants from environmental influences, sedimentary facies analysis is crucial before selecting plant samples for carbon isotope testing.

Over the past decade, we have conducted sedimentary surveys in South China with the assistance of numerous collaborators. We collected plant fossils from various sedimentary facies and reconstructed plant habitats based on fossil preservation conditions and sedimentary facies[5]. Our study covers floras collected from terrestrial, paralic and deep-sea facies[5]. To assess the impact of atmospheric $CO_2$ pressure, we sampled plant fossils with carbon films or cuticles from the Late Permian to the Middle Triassic, alongside proxy-based atmospheric $CO_2$ content reconstructions for each substage. The age, facies and palaeoenvironments of each flora are detailed in ref. 5.

The specific parts of the plant fossils that were sampled are detailed in the Supplementary Information.

To ensure that the carbon isotope samples are derived from plant fossils, we analysed both the organic matter from the plant fossils and the surrounding rock matrix. An -1‰ difference in $\delta^{13}C_{org}$ between the plant fossils and the surrounding matrix confirms the reliability of the samples[54]. Matrix samples were cleaned with compressed air to prevent cross-contamination and are documented in the Supplementary Information. Only identifiable plant fossils were sampled. Samples were extracted using an alloy scalpel, with a minimum of 20 mg per plant body fossil and 5 g for surrounding rock. To avoid contamination by surrounding matrix to the plant fossil samples, we systematically scratched as thin layers as possible. To get enough sampling amount for small plants, for example, the Triassic lycopods, some samples of the same species and specimen were gathered as one sample, resulting in fewer samples but higher accuracy of each datapoint. Considering each part of the plants may bear slightly different carbon isotope value, all parts of each plant fossil were sampled, including leaves (vegetative/sporophylls), branches, seeds, petioles and veins.

To eliminate the influence of inorganic carbon on the carbon isotope signal, all samples for organic carbon isotope testing—including plant carbon such as cuticles and surrounding rock—were treated with 15% HCl acid then repeatedly rinsed with deionized water before drying at 45 °C and subsequent crushing. The description of each sample and the carbon isotope data are presented in the Supplementary Information. The prepared samples were analysed for organic carbon isotope ratios using a Mat253 Plus (Thermo Fisher, MAT 253 Plus Isotope Ratio Mass Spectrometer) and a Delta V advantage (Thermo Fisher) at the State Key Laboratory of Biogeology and Environmental Geology, China University of Geosciences (Wuhan), and an EA-IRMS system (Elemental Analyzer–Isotope Ratio Mass Spectrometry) at the Stable Isotope Facility, Department of Plant Sciences, University of California, Davis. For the Mat253 Plus, calibration was based on GBW (Guobiao Wuzhi, Chinese National Standard Reference Materials) standards (GBW04407, −22.43; GBW04408, −36.91‰) with ACET (acetanilide) (−26.33‰) as the internal standard. For the Delta V Advantage, reference materials included USGS40 (−26.39‰), USGS24 (−16.05‰), and IVA33802174 Urea (−37.32‰). For the EA-IRMS system, multiple laboratory reference materials were used for scale normalization and quality control, including caffeine ($\delta^{13}C$ −34.90 ± 0.09‰; $\delta^{15}N$ −2.74 ± 0.10‰), glutamic acid ($\delta^{13}C$ −10.98 ± 0.10‰; $\delta^{15}N$ −8.54 ± 0.08‰), glutathione ($\delta^{13}C$ −18.27 ± 0.07‰; $\delta^{15}N$ −5.00 ± 0.04‰), scallop ($\delta^{13}C$ −16.74 ± 0.10‰; $\delta^{15}N$ 9.37 ± 0.06‰) and nylon powder ($\delta^{13}C$ −24.90 ± 0.05‰; $\delta^{15}N$ −1.12 ± 0.16‰), among others. Analytical precision was better than ±0.1‰ ($1\sigma$) for standards and typically within ±0.2‰ for samples, with a maximum uncertainty of ±0.5‰ in cases of low signal intensity or abnormal matrices (for example, high halogen or sulfur contents). Replicate analyses of samples yielded reproducibility better than ±0.2‰ (Supplementary Table 5). All remaining samples and plant fossils are stored in Room 014B, Main Building, China University of Geosciences (Wuhan) and University of Leeds.

## HadCM3BL model simulations

HadCM3BL is an Earth system model that incorporates atmosphere, ocean, land and biosphere, developed by the UK Metoffice and University of Bristol[3]. Specifically, we use HadCM3LB-M2.1aD with a grid resolution of 3.75° × 2.5° in longitude × latitude in both the atmosphere (19 vertical levels) and ocean (20 vertical levels), using the Arakawa B-grid scheme. The model uses a dynamic vegetation scheme, which is crucial for such studies: the Top-Down Representation of Interactive Foliage and Flora Including Dynamics with the MOESE 2.1 land surface scheme. Desert soil albedo is interactively updated on the basis of the soil carbon content, where low soil carbon concentrations result in a modified soil albedo of 0.32 (average modern-day Saharan albedo).

Typically, the ozone distribution is prescribed as a static latitude–pressure–time distribution in many climate models. However, as the climate warms, the tropopause rises, meaning that stratospheric ozone penetrates into the troposphere, which is unphysical if a pre-industrial tropopause height is prescribed for warm time periods. Instead, the ozone distribution is prescribed using a dynamic approach in which ozone is dynamically coupled to the model tropopause height with constant values for the troposphere (0.02 p.p.m.), tropopause (0.2 p.p.m.) and stratosphere (5.5 p.p.m.). This change makes a negligible difference to the global mean surface temperature but does have a small impact on the stratospheric temperature and winds.

A range of boundary conditions are required to configure the model for Permo-Triassic conditions. The Getech Plc. palaeogeography (land–sea distribution, bathymetry, topography) is used as well as time-specific atmospheric $p$CO$_2$ (detailed below) and solar luminosity. Each simulation was fully equilibrated in both the atmosphere and deep ocean following a three-stage spin-up protocol so that each simulation is fully equilibrated: (1) the globally and volume-integrated annual mean ocean temperature trend is less than 1 °C per 1,000 years, (2) trends in surface air temperature are less than 0.3 °C per 1,000 years, and (3) net energy balance at the top of the atmosphere, averaged over a 100 year period at the end of the simulation, is less than 0.25 W m$^{-2}$. These simulations have generally been run for over 10,000 model years to ensure complete Earth system equilibrium. Climate means were then produced from the last 100 years of the simulation.

Using systematic proxy data, including sea surface temperature, atmospheric CO$_2$ and sedimentary observations such as climatically sensitive minerals/facies, HadCM3BL successfully established robust simulations across the PTME interval that shows a mega-El Niño and stronger temperature fluctuations both on land and in the ocean due to the collapse of meridional overturning circulation and a contracted Hadley cell[3].

In this work, we ran the end-Permian Changhsingian, PTT and Early Triassic Induan scenarios using HadCM3BL with atmospheric CO$_2$ concentrations of 412 p.p.m., 2,568 p.p.m. and 4,000 p.p.m., respectively, derived from boundary values reconstructed using plant stomatal, palaeosol and plant carbon isotope fractionation proxies[4,17,18]. All simulations can be found on the Bristol BRIDGE website (https://www.bristol.ac.uk/geography/research/bridge/). After stabilization of the atmosphere–ocean–vegetation coupling, we outputted the average and absolute maximum daily land surface temperature.

The global average maximum daily land surface temperature is the average of each day's highest temperature over a year, describing the overall thermal intensity experienced by the land surface[74]. This metric is essential in capturing the cumulative effect of heat extremes, which are critical for assessing the habitability of terrestrial environments, especially for vegetation. Unlike mean annual temperature, this index reflects both the frequency and intensity of high-temperature events, providing insights into seasonal thermal stress and potential physiological thresholds for plant survival and function[74].

The global absolute maximum daily land surface temperature, by contrast, captures the single highest temperature recorded in each grid cell of each scenario. This metric reflects the most extreme thermal event experienced at each location, providing crucial information on the upper thermal limits of the environment. It is particularly valuable for evaluating the survivability of organisms under short-term extreme heat stress, which may exceed physiological thresholds even if average conditions are tolerable. This parameter helps identify thermal hotspots and assess the risks of episodic temperature extremes that can drive ecological collapse or restrict species distributions.

## Reporting summary

Further information on research design is available in the Nature Portfolio Reporting Summary linked to this article.

## Data availability

The data that support the findings of this study are included in the paper and/or the Supplementary Information and Supplementary Data 1.

## Code availability

The NNA analysis code is available in ref. 72. The HadCM3BL climate model is available in ref. 3. The R code used for parts of the PCA analysis is provided in the Supplementary Information.

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

## Acknowledgements

We thank H. F. Yin, F. S. Meng, W. J. Ran, Q. Xue, Y. H. Guo, X. Shi, W. C. Shu, L. Zhang, Y. Y. Tian, X. J. Wang, M. J. Zhang, G. Z. Xu, B. B. Li, M. Fan and W. J. Lin for fieldwork assistance and H. F. Yin, J. Y. Wan, A. H. Yuan, S. Z. Gu and M. H. Zhang for help with clustering methods and for discussion of results. A. Spencer is thanked for help with neighbourhood network methods and E. Kustatscher, D. Royer and R. Bateman for thoughtful discussion on all aspects of this work. We thank Q. P. He for making the reference searching program. G. M. Luo, X. Y. Ma, B. Chang, K. P. Ewert, X. Q. Zhang, D. D. Li, H. Zhao, L. S. Zhao, L. Zhang, Y. Du and H. Y. Song are also thanked for assistance with carbon isotopes experiments. This work is financially supported by the National Natural Science Foundation of China (grant 42430209) (J.Y., Z.X., N.P.), UK Research and Innovation project EP/Y008790/1

(Z.X., B.J.W.M.), Natural Environment Research Council NE/T000392/1 (B.H.L.), Human Frontiers Science Program grant RGP0066/2021 (B.H.L.), the UK Palaeontological Association Sylvester-Bradley Award PA-SB202406 (Z.X.) and the US National Science Foundation (FRES 2121594) (I.P.M., Z.X.). We thank Y. Chi for colouring the reconstructions in the figures.

## Author contributions

Z.X., J.Y., J.H. and B.H.L. designed the study. Z.X., J.H., B.H.L., J.Y. and X.S. developed the methods. Z.X., N.P. and J.Y. undertook the field work. Z.X. and N.P. conducted laboratory experiments. A.F. ran the climate simulations. Q.L. developed the data searching code. Z.X., J.H., B.H.L., B.J.W.M., P.B.W., I.P.M., A.F. and J.Y. analysed the data and made the figures. Z.X., B.H.L. and J.H. wrote the first draft with contributions from B.J.W.M., P.B.W., I.P.M., A.F. and J.Y.

## Competing interests

The authors declare no competing interest.

## Additional information

**Correspondence and requests for materials** should be addressed to Zhen Xu or Jianxin Yu.

# Reporting Summary

## Statistics

For all statistical analyses, confirm that the following items are present in the figure legend, table legend, main text, or Methods section.

| n/a | Confirmed | |
|---|---|---|
| ☐ | ☒ | The exact sample size (*n*) for each experimental group/condition, given as a discrete number and unit of measurement |
| ☐ | ☒ | A statement on whether measurements were taken from distinct samples or whether the same sample was measured repeatedly |
| ☒ | ☐ | The statistical test(s) used AND whether they are one- or two-sided<br>*Only common tests should be described solely by name; describe more complex techniques in the Methods section.* |
| ☒ | ☐ | A description of all covariates tested |
| ☒ | ☐ | A description of any assumptions or corrections, such as tests of normality and adjustment for multiple comparisons |
| ☒ | ☐ | A full description of the statistical parameters including central tendency (e.g. means) or other basic estimates (e.g. regression coefficient) AND variation (e.g. standard deviation) or associated estimates of uncertainty (e.g. confidence intervals) |
| ☒ | ☐ | For null hypothesis testing, the test statistic (e.g. *F*, *t*, *r*) with confidence intervals, effect sizes, degrees of freedom and *P* value noted<br>*Give P values as exact values whenever suitable.* |
| ☒ | ☐ | For Bayesian analysis, information on the choice of priors and Markov chain Monte Carlo settings |
| ☒ | ☐ | For hierarchical and complex designs, identification of the appropriate level for tests and full reporting of outcomes |
| ☒ | ☐ | Estimates of effect sizes (e.g. Cohen's *d*, Pearson's *r*), indicating how they were calculated |

*Our web collection on statistics for biologists contains articles on many of the points above.*

## Software and code

Policy information about availability of computer code

| Data collection | Open access software: ImageJ |
|---|---|
| Data analysis | Open access software: PAST, Mesquite, PAUP, SpiltTree, Matlab, R |

For manuscripts utilizing custom algorithms or software that are central to the research but not yet described in published literature, software must be made available to editors and reviewers. We strongly encourage code deposition in a community repository (e.g. GitHub). See the Nature Portfolio guidelines for submitting code & software for further information.

## Data

Policy information about availability of data

All manuscripts must include a data availability statement. This statement should provide the following information, where applicable:

- Accession codes, unique identifiers, or web links for publicly available datasets
- A description of any restrictions on data availability
- For clinical datasets or third party data, please ensure that the statement adheres to our policy

The data that support the findings of this study are included in the paper and/or the supplementary information.

# Research involving human participants, their data, or biological material

Policy information about studies with [human participants or human data](). See also policy information about [sex, gender (identity/presentation), and sexual orientation]() and [race, ethnicity and racism]().

| | |
|---|---|
| Reporting on sex and gender | *Use the terms sex (biological attribute) and gender (shaped by social and cultural circumstances) carefully in order to avoid confusing both terms. Indicate if findings apply to only one sex or gender; describe whether sex and gender were considered in study design; whether sex and/or gender was determined based on self-reporting or assigned and methods used.*<br>*Provide in the source data disaggregated sex and gender data, where this information has been collected, and if consent has been obtained for sharing of individual-level data; provide overall numbers in this Reporting Summary. Please state if this information has not been collected.*<br>*Report sex- and gender-based analyses where performed, justify reasons for lack of sex- and gender-based analysis.* |
| Reporting on race, ethnicity, or other socially relevant groupings | *Please specify the socially constructed or socially relevant categorization variable(s) used in your manuscript and explain why they were used. Please note that such variables should not be used as proxies for other socially constructed/relevant variables (for example, race or ethnicity should not be used as a proxy for socioeconomic status).*<br>*Provide clear definitions of the relevant terms used, how they were provided (by the participants/respondents, the researchers, or third parties), and the method(s) used to classify people into the different categories (e.g. self-report, census or administrative data, social media data, etc.)*<br>*Please provide details about how you controlled for confounding variables in your analyses.* |
| Population characteristics | *Describe the covariate-relevant population characteristics of the human research participants (e.g. age, genotypic information, past and current diagnosis and treatment categories). If you filled out the behavioural & social sciences study design questions and have nothing to add here, write "See above."* |
| Recruitment | *Describe how participants were recruited. Outline any potential self-selection bias or other biases that may be present and how these are likely to impact results.* |
| Ethics oversight | *Identify the organization(s) that approved the study protocol.* |

Note that full information on the approval of the study protocol must also be provided in the manuscript.

# Field-specific reporting

Please select the one below that is the best fit for your research. If you are not sure, read the appropriate sections before making your selection.

☐ Life sciences    ☐ Behavioural & social sciences    ☒ Ecological, evolutionary & environmental sciences

For a reference copy of the document with all sections, see [nature.com/documents/nr-reporting-summary-flat.pdf]()

# Ecological, evolutionary & environmental sciences study design

All studies must disclose on these points even when the disclosure is negative.

| | |
|---|---|
| Study description | This study investigates the survival strategies of lycophyte plants across the Permian–Triassic Mass Extinction (PTME), with a focus on whether Crassulacean Acid Metabolism (CAM) photosynthesis enabled their persistence under extreme environmental stress. The study combines phylogenetic analysis, morphological quantification (127 traits), carbon isotope geochemistry, and Earth system climate modeling (HadCM3BL). It uses a factorial design integrating multiple methods across time intervals (Late Permian to Middle Triassic) and geographic regions. |
| Research sample | The research sample consists of 485 lycophyte sporophyll fossil specimens from the Late Permian to Middle Triassic of South China and other global sites, along with 200 literature-derived fossil data points. The focus is on sporophylls, as these reproductive structures preserve the most diagnostic characters. The sample includes both extinct taxa and extant Isoetes species to assess phylogenetic and ecophysiological continuity. Specimens were collected from fieldwork and literature and were selected based on preservation quality and stratigraphic context. |
| Sampling strategy | No formal statistical power calculation was performed due to fossil record constraints. Sample sizes were determined based on fossil availability and completeness, with efforts to maximize taxonomic, stratigraphic, and geographic coverage. Sampling aimed to include all morphologically character-rich sporophylls suitable for morphometric and isotopic analysis. Carbon isotope samples pooled material from the same species, locality, and stratigraphic level to ensure data quality. |
| Data collection | Morphological data were scored from fossils collected in this work and also references using a 127-character matrix and visualized via PCA and Neighborhood Network. Analysis Isotope data were collected from the cuticle-bearing surfaces of plant fossils collected using a scalpel, avoiding matrix contamination. All the fossils are cleaned using compressed before sampling. Data were recorded and processed by the lead author and collaborators at the University of Leeds and China University of Geosciences Wuhan. Climate model simulations were run by a co-author using HadCM3BL under different $CO_2$ scenarios, which have been previously published. |
| Timing and spatial scale | Fossil data span ~254–237 Ma, covering the Changhsingian, Permian–Triassic transition, and Early–Middle Triassic. Fossils used for phylogeny were collected from multiple sedimentary facies across South China, in field work with additional data from North China, China Xinjiang, Europe, Australia, Russia, middle Asia, Antarctica, Argentina in references. The carbon isotope data only come from |

| | |
|---|---|
| | South China because South China was in the low latitude tropical area thus has the highest temperature and highest potential of CAM. The climate simulations are come from a published paper (Sun et al., 2024, Science), and the full simulations can be found in the Bristol BRIDGE website (https://www.bristol.ac.uk/geography/research/bridge/). |
| Data exclusions | Fossils lacking key morphological features or too poorly preserved for confident character coding were excluded. Similarly, literature-derived isotope data without associated sediment $\delta^{13}C$ values were excluded from the core analysis. The rationale for all exclusions is described in the Methods and Supplementary Information. No exclusions were based on results. |
| Reproducibility | All data, including morphological matrices, specimen photographs, images of carbon isotope sampling sites on both fossils and rocks, and measured $\delta^{13}C$ values, are provided in the Supplementary Information. PCA and NNA procedures are described in full detail with annotated figures, and all software used (PAST, Mesquite, PAUP, SplitTree, and R) is freely available. These methods can therefore be readily applied to other fossil datasets. Carbon isotope measurements were independently conducted at the China University of Geosciences (Wuhan) and the University of California, Davis, using three different sets of instruments and standards, yielding consistent results. Remaining isotope samples are archived at the University of Leeds. Climate model simulations are reproducible using the HadCM3BL code and settings described in the cited references, and code availability is provided. For figures containing small symbols that cannot be enlarged due to journal size constraints, we have included a statement noting that all source data and code are available in the Supplementary Information, enabling others to reproduce the analyses and generate high-resolution vector figures. |
| Randomization | Randomization is not applicable, as this is a retrospective fossil-based study. Sample selection was determined by preservation quality, stratigraphic control, and specimen completeness, not experimental manipulation. Covariates such as geography, facies, and age were recorded for all samples and considered in interpretation. |
| Blinding | Blinding was not relevant to this study. Data acquisition and analysis involved fossil specimens and geochemical measurements that are not subject to observer bias in the conventional sense. Morphological character scoring followed predefined criteria, and isotopic analyses were performed using standardized laboratory protocols. Sample selection was based on preservation quality and stratigraphic control, not experimental grouping, making blinding unnecessary. |

Did the study involve field work? ☒ Yes ☐ No

# Field work, collection and transport

| | |
|---|---|
| Field conditions | The field works are conducted during summer in South China when the highest day temperature is below 35 degree and we only work when there is no rain to avoid potential risks. |
| Location | The field works are in Hubei, Hunan, Guizhou, Yunnan, Sichuan provinces in South China. The carbon isotope experiments are taken in the China University of Geosciences Wuhan and University of California Davis. |
| Access & import/export | Fossil sampling was authorized by an official introduction letter from China University of Geosciences (Wuhan), which permitted access to fossil localities and collection in compliance with Chinese national regulations. The carbon isotope samples transported to University of Californian Davis were accompanied by an official academic invitation letter and a written explanation of research purpose. |
| Disturbance | This study caused minimal disturbance. All specimens were collected from previously excavated or naturally exposed outcrops, avoiding any protected or ecologically sensitive areas. No living ecosystems or habitats were altered. Sampling was conducted by hand without the use of heavy machinery. |

# Reporting for specific materials, systems and methods

We require information from authors about some types of materials, experimental systems and methods used in many studies. Here, indicate whether each material, system or method listed is relevant to your study. If you are not sure if a list item applies to your research, read the appropriate section before selecting a response.

## Materials & experimental systems

| n/a | Involved in the study |
|---|---|
| ☒ | ☐ Antibodies |
| ☒ | ☐ Eukaryotic cell lines |
| ☐ | ☒ Palaeontology and archaeology |
| ☒ | ☐ Animals and other organisms |
| ☒ | ☐ Clinical data |
| ☒ | ☐ Dual use research of concern |
| ☒ | ☐ Plants |

## Methods

| n/a | Involved in the study |
|---|---|
| ☒ | ☐ ChIP-seq |
| ☒ | ☐ Flow cytometry |
| ☒ | ☐ MRI-based neuroimaging |

# Palaeontology and Archaeology

Specimen provenance
Fossil specimens were collected under an official introduction letter issued by China University of Geosciences (Wuhan), which permitted access and collection at designated fossil localities across Hubei, Hunan, Guizhou, Yunnan, and Sichuan provinces in China. Carbon isotope samples were transported to the University of California, Davis under an academic invitation and collaboration agreement for non-commercial scientific research. All specimens were collected in accordance with institutional and national regulations at the time of sampling.

Specimen deposition
All specimens and remaining samples are permanently stored at the China University of Geosciences (Wuhan), Room 014B of the Main Building, and at the University of Leeds, School of Earth and Environment. These repositories allow access to qualified researchers upon request.

Dating methods
No new radiometric or calibrated dates were produced in this study. Stratigraphic age assignments for the fossil specimens are based on previously published biostratigraphy and lithostratigraphy of the Kayitou, Feixianguan, and Badong formations. Full details and references are provided in the Supplementary Information and main text.

☒ Tick this box to confirm that the raw and calibrated dates are available in the paper or in Supplementary Information.

Ethics oversight
No ethical approval was required for this study, as it did not involve living organisms or human subjects. All fossil material was collected and studied in accordance with institutional and national guidelines for palaeontological research.

Note that full information on the approval of the study protocol must also be provided in the manuscript.

# Plants

Seed stocks
*Report on the source of all seed stocks or other plant material used. If applicable, state the seed stock centre and catalogue number. If plant specimens were collected from the field, describe the collection location, date and sampling procedures.*

Novel plant genotypes
*Describe the methods by which all novel plant genotypes were produced. This includes those generated by transgenic approaches, gene editing, chemical/radiation-based mutagenesis and hybridization. For transgenic lines, describe the transformation method, the number of independent lines analyzed and the generation upon which experiments were performed. For gene-edited lines, describe the editor used, the endogenous sequence targeted for editing, the targeting guide RNA sequence (if applicable) and how the editor was applied.*

Authentication
*Describe any authentication procedures for each seed stock used or novel genotype generated. Describe any experiments used to assess the effect of a mutation and, where applicable, how potential secondary effects (e.g. second site T-DNA insertions, mosiacism, off-target gene editing) were examined.*

