## [Peer Review File · Nature Ecology & Evolution]

CAM photosynthesis may have conferred an advantage during the Permian-Triassic mass extinction event

Corresponding Author: Dr ZHEN XU

Version 0:

Decision Letter:

5th August 2025

Dear Dr Xu,

Your manuscript entitled "CAM photosynthesis: a key trait in surviving Earth's largest extinction" has now been seen by three reviewers, whose comments are attached. The reviewers have raised a number of concerns which will need to be addressed before we can offer publication in Nature Ecology & Evolution. We will therefore need to see your responses to the criticisms raised and to some editorial concerns, along with a revised manuscript, before we can reach a final decision regarding publication.

We therefore invite you to revise your manuscript taking into account all reviewer and editor comments. Please highlight all changes in the manuscript text file [OPTIONAL: in Microsoft Word format].

* If you have not done so already please begin to revise your manuscript so that it conforms to our Article format instructions at <http://www.nature.com/natecolevol/info/final-submission>. Refer also to any guidelines provided in this letter.

* Extended Data Figures - please ensure that any supplementary figures and tables that are crucial to the manuscript's conclusions are converted into Extended Data figures and tables to increase visibility of these data. Extended Data figures and tables are online-only (present in the online PDF and full-text HTML versions of the paper), peer-reviewed display items that provide essential background to the article but are not included in the main article due to space constraints. A maximum of ten Extended Data display items (figures and tables) is permitted.

Link Redacted

Nature Ecology & Evolution is committed to improving transparency in authorship. As part of our efforts in this direction, we are now requesting that all authors identified as 'corresponding author' on published papers create and link their Open Researcher

and Contributor Identifier (ORCID) with their account on the Manuscript Tracking System (MTS), prior to acceptance. ORCID helps the scientific community achieve unambiguous attribution of all scholarly contributions. You can create and link your ORCID from the home page of the MTS by clicking on 'Modify my Springer Nature account'. For more information please visit www.springernature.com/orcid.

[redacted]

Reviewer expertise:

Reviewer #1: carbon cycle, geochemistry

Reviewer #2: isotope geochemistry

Reviewer #3: signed report

Reviewers' comments:

Reviewer #1 (Remarks to the Author):

The authors investigated the morphology and carbon isotopic composition of lycophyte sporophylls across the Permian-Triassic mass extinction event and found evidence that lycophytes during and shortly after the extinction were particularly adapted to hot-house conditions. This proposition is supported by a climate model that reconstructs paleotemperatures in the studied habitats. The authors conclude that this physiological adaptation may have enabled the terrestrial biosphere to survive the prolonged hot-house event.

I emphasize that I am not a paleobotanist and can therefore not comment on those aspects of the manuscript. My comments will focus on aspects of isotope geochemistry and Earth system science.

I find the results overall intriguing and impressive. The carbon isotope study is done carefully, with appropriate comparison to background biomass and other plants within the same sample. The interpretation of this dataset as evidence of CAM-like photosynthesis is compelling (although I stress that I cannot comment on the plausibility of CAM photosynthesis in these particular organisms at this time, because that is outside of my field). To an outsider, the paleobotanic dataset looks convincing. However, I wondered if there is any risk that the conclusions may be skewed by the fact that the new morphology-based taxonomy lacks resolution (as noted in the manuscript). Is there any risk that ecophysiological information could be misinterpreted (see also comment below)?

Regarding the overall conclusions about broader implications for survival of the biosphere and feedbacks on global climate, it would be helpful if information could be provided on what fraction of the terrestrial flora was composed of CAM-photosynthesizers at any given time. The discussion states that these were dominant and controlling carbon burial globally, but the $\delta^{13}\text{C}$ data imply that other non-CAM plants were present. A mass balance calculation would be helpful to quantitatively support this broader conclusion and the title of the manuscript.

Line comments:

I. 31 and 49: the word 'dramatic' has no scientific meaning. You can delete it without loss of content.

I. 116: data are

II. 231-244: I appreciate that the authors acknowledge the limitations of their approach. However, it leaves me wondering if the loss of taxonomic resolution could lead to false interpretations about the ecophysiology of these organisms. In other words, could sporophytes from organisms adapted to distinct environmental conditions still look identical, based purely on morphological characteristics? And so, could it happen that these fossils are misinterpreted?

LI. 252-254: coastal lowlands can be very heterogeneous in salinity, as they are located along the interface between freshwater and seawater. Can additional evidence be provided that salinity is not a concern for this analysis?

I. 269: delete the comma

I. 274: data indicate (the word 'data' is plural)

I. 277: change to "...share a similar carbon isotope composition of circa..."

I. 409: Maybe specify what the non-lycophytes are, so that the reader can be more certain that they could not have been performing CAM-based photosynthesis. I am not familiar with how widespread this pathway is among plants and if it could be present in non-lycophytes of that age. Other readers may have the same question.

I. 446: is 'dominated by CAM plants' the correct phrase to use? After all, there are still many non-lycophytes in the samples that do not express CAM photosynthesis (as evidenced by the isotopic data).

I. 445: related to my previous comment: Terrestrial carbon burial would only change significantly if CAM plants made up a large fraction of the terrestrial flora. This point is important to resolve. Perhaps an isotopic mass balance calculation could be applied to the samples to determine what fraction of the terrestrial biomass was CAM-derived?

I. 462: Also here, it needs to be shown first that these organisms were indeed the major remnants, in comparison to other plants.

II. 462-465: This sentence is not entirely clear. Why would the absence of CAM plants have extended the hothouse conditions beyond 5 million years? This point seems to contradict the statement in II. 454-456, which mentioned a positive feedback that increased warming. Please clarify.

Fig. 5: It would be helpful to add ages in numbers to make this figure easier to read for people who are not familiar with the stage names. It would make it easier to compare to Fig. 4.

II. 676-679: Please provide information about analytical reference materials and report precision and accuracy of the data.

Eva Stüeken

Reviewer #2 (Remarks to the Author):

This is a thought-provoking study. Xu et al. use lycophyte sporophyll morphology to assemble a new phylogeny and argue that the highly prevalent lycophytes in the aftermath of the Permian-Triassic mass extinction had a survival advantage due to employing CAM photosynthesis. They bolster the interpretations of their morphological trait-based phylogeny with carbon isotope data from fossil lycophytes showing diminished carbon isotope fractionation (consistent with CAM) during the PTME, as well as climate model simulations showing prohibitively high temperatures for C₃ photosynthesis.

Overall I think this is a neat set of observations that is tied together with a plausible explanation. The field would benefit from seeing this in the literature. That said, many details are not (and cannot be) fully sorted out. I think the paper works well as a "hypothesis paper", rather than a definitive assessment of the photosynthetic metabolism of fossil lycophytes and its role in lycophyte (or broader biosphere) survival (to that end, my first suggestion is to end the title with a question mark!). I have some suggestions below to open up a bit more discussion on points of contention, and with attention to those I think this would be acceptable for publication.

I am less familiar with the basis for the morphological trait analyses, so my comments pertain to the carbon isotopes and plant physiology:

1. Carbon isotopes and the inference of CAM. While C isotopes are a crucial piece of evidence for inferring CAM activity in deep time, they alone can't be definitive, for a few reasons. First, the C isotope effects resulting from CAM overlap the lower range of those generated by C₃ photosynthesis. In terms of carbon mass balance, that end-member of C₃ photosynthesis is similar to CAM (being more efficient at C fixation). But metabolically, inferring CAM means the plants are employing organic acid accumulation with temporal separation of C uptake & fixation, whereas the low-D¹³C end-member of C₃ photosynthesis does not. Second, we don't precisely know the D¹³C values, because we don't precisely know the d¹³C_CO₂ value through the event. So the absolute value of D¹³C cannot be used to distinguish CAM vs. C₃. In light of this, the authors compare d¹³C of lycophytes to non-lycophytes through the PTME. This is sensible, but non-lycophytes sample sizes are small, particularly for the "transition" phase. Third, even if the n=3 sampling of non-lycophytes captures an accurate value for the "transition" phase, it is possible that these outgroup plants respond differently than lycophytes to CO₂, moisture or insolation, such that both groups moved to lower d¹³C due to the drop in d¹³C_CO₂, but the non-lycophytes increased their D¹³C (e.g., due to high CO₂) while lycophytes remained the same. In such a scenario, there would be no reason to infer CAM in the lycophytes. As a final note, I appreciate the discussion of aqueous CO₂ uptake, but I don't think the isotopic data can give a clear vote for or against it (very uncertain what the d¹³C of that CO₂ source would be, depending on amount of sedimentary respired carbon vs. marine DIC). I mention these issues not necessarily to say I think it is unlikely that the lycophytes employed CAM, but rather to say the available data are not definitive.

2. Temperature limits of extinct C₃ plants and CAM as a survival mechanism. The statement in Line 330: "The survival of these plants under such extreme heat [45-65 C] strongly suggests the use of an alternative photosynthetic pathway, rather than the C₃ type." needs more unpacking. First, while 45-65C is indeed terribly hot, the cited paper (ref. 19) shows that CAM plants have lower optimal growth temperatures than C₃, with only C₄ having higher temperature tolerance. On this basis, it would seem that CAM does not help explain lycophyte survival. Second, as noted in Line 322, extant plant lineages (not just C₃, but CAM as well) evolved long after the interval being studied here. The applicability of modern experiments (as in ref. 19) to these ancestral lineages is difficult to prove. There's no way around that, but it remains an inherent limitation of this sort of study. Third, if higher temperature did indeed favor CAM, please explain the mechanism in more metabolic detail. The additional sentences about lycophyte survival in the Discussion (lines 459-462) don't add much to that, and also bring in tangential ideas such as "anti-heavy metal antioxidant enzyme systems" that are not clearly tied to the environmental perturbations being discussed. Overall, the role of CAM in lycophyte survival needs better explaining.

Reviewer #3 (Remarks to the Author):

Zhen Xu et al present an important, data rich and multi-faceted analysis on the taxonomy, ecophysiology and climatic / biogeographic distribution of fossil herbaceous lycophytes from before, during and after the Permian-Triassic mass extinction event. The study brings fresh insights, new analyses and a multi-disciplinary approach to furthering understanding of plant species resilience to extreme climatic warmth during this time interval. Xu et al propose based on detailed morphological character analysis of 485 fossil lycophyte specimens that the Permian-Triassic transition interval is characterised by a taxon of herbaceous lycophytes which are morphologically (and taxonomically) and ecophysiological distinct from their ancestors and descendant clades. The author team argue that the P-T transition fossil likely possessed CAM photosynthesis similar to the living quillworts (Isoetids) based on having less negative isotopic value compared with contemporaneous non-lycophyte taxa and shared morphological traits. They go on to argue that CAM photosynthesis which has a higher daytime photosynthetic temperature tolerance than C3 taxa and much higher water use efficiencies would have conferred resilience to the surviving herbaceous lycophytes in the transition interval.

Overall the study is exciting, novel and robust and brings a wealth of new fossil data and analyses to the Permian-Triassic mass extinction interval. Although the idea of flexible photosynthesis as a strategy to withstand earth extreme events is not new in itself (see papers by Looe, Vischer, Green etc- all of which are cited in the manuscript) – the careful and detailed analysis of Lycophyte sporophylls as a primary source of data is highly novel.

I think the paper can be further strengthened with a few additional considerations by the author team as follows:

- (1) Throughout the manuscript the authors talk about switching from C3 to CAM. The overall suggestion throughout the manuscript is that the PT transition taxa are obligate CAM? yet it is not stated if the authors think they are obligate or facultative CAM. Looking at the carbon isotope data I think it is equally parsimonious to suggest that the plants are facultative CAM – they carry out C3 photosynthesis predominantly but can switch within their lifetime to CAM photosynthesis under extreme conditions (eg submergence or high temperature/ high aridity stress). A subtle tightening up of the language should resolve this uncertainty.
- (2) Taphonomy and differences in depositional environment are somewhat glossed over in the PCA analysis. Could an additional set of analyses be included which codes the fossil taxa in the PCA morphospace with taphonomic or depositional environment/ facies ? This would help to test if the groupings within the PCA are indeed due to morphological similarities and differences due to evolutionary differences rather than local environmental signals? I don't think there is an issue here but it would be nice to conduct this test if it is possible.
- (3) There seems to be a small inconsistency in the CO2 trends described in the intro and those used for the HAD gcm analyses – please double check.
- (4) I have made detailed comments and suggestions throughout the attached manuscript file with all my additional minor comments to improve the manuscript.

In summary, this is a novel, exciting and data rich paper that combines taxonomic , phylogenetic, paleoecophysiological and paleogeographic and paleoclimate analyses to make a strong case that the disaster herbaceous taxa which were present during the Permian-Triassic transition interval are phylogenetically distinct from their Permian ancestors and possessed novel photosynthetic biology that enabled them to survive one of the greatest extinction events in Earth history.

Jenny McElwain

*****END*****

Version 1:

Decision Letter:

18th December 2025

Dear Dr. Xu,

Thank you for submitting your revised manuscript "CAM photosynthesis: a key trait in surviving Earth's largest extinction?" (NATECOLEVOL-25072164A). It has now been seen again by the original reviewers and their comments are below. The reviewers find that the paper has improved in revision, and therefore we'll be happy in principle to publish it in Nature Ecology & Evolution, pending minor revisions to satisfy the reviewers' final requests and to comply with our editorial and formatting guidelines.

If you have not done so already, please ensure that you also email us a completed copy of the Reporting summary :

Reporting summary: https://www.nature.com/documents/nr-reporting-summary.pdf

We are now performing detailed checks on your paper and will send you a checklist detailing our editorial and formatting

requirements in about a week (note that due to our end of year closures unfortunately this could take a bit longer). Please do not upload the final materials and make any revisions until you receive this additional information from us.

[redacted]

Reviewer #1 (Remarks to the Author):

I read through the author's replies and the revised manuscript, and I think my previous comments have been addressed well. I don't have any additional questions and look forward to seeing the paper in press.

Best wishes,
Eva Stüeken

Reviewer #1 (Remarks on code availability):

The code is very complex and beyond my area of expertise. However, it is based on previously published modelling frameworks. I therefore trust that it is reliable.

Reviewer #2 (Remarks to the Author):

The authors have adequately addressed my comments and those of the other reviewers. I am happy to see this version of the manuscript published.

Reviewer #3 (Remarks to the Author):

It is a pleasure to re-evaluate this manuscript on 'Cam photosynthesis, a key trait in surviving Earth's largest extinction'. I have read the detailed responses to reviewers and the revised manuscript and feel that the authors have thoroughly addressed my initial concerns and those of the other reviewers. The tone of the manuscript is very well pitched following the reviewer comments highlighting the advances the paper makes in relation to the evidence for CAM and also avenues for further testing of the CAM hypothesis. I think this is a really important paper and makes a significant advance to our understanding of how vegetation can withstand extreme intervals in Earth history. It also elegantly demonstrates the importance of paleobiology in understanding past and current earth system processes which are becoming increasingly relevant as the Earth transitions from a coldhouse to coolhouse climate state.

I have one remaining minor suggestion:

Line 351 the authors state that 'Alternatively, CAM photosynthesis has been previously hypothesized in deeper time^{12,38,47,48}

Please add two further citations to support this statement as both have suggested earlier evolution of CAM and ways of detecting it in the fossil record.

Raven JA, Spicer RA. The evolution of crassulacean acid metabolism. In Crassulacean acid metabolism: biochemistry, ecophysiology and evolution 1996 (pp. 360-385). Berlin, Heidelberg: Springer Berlin Heidelberg.

McElwain JC, Mattheus WJ, Barbosa C, Chondrogiannis C, O'Dea K, Jackson B, Knetge AB, Kwasniewska K, Nair R, White JD, Wilson JP. Functional traits of fossil plants. New Phytologist. 2024 Apr;242(2):392-423.

Version 2:

Decision Letter:

18th February 2026

Dear Dr XU,

We are pleased to inform you that your Article entitled "CAM photosynthesis may have conferred an advantage during the Permian-Triassic mass extinction event", has now been accepted for publication in Nature Ecology & Evolution.

Over the next few weeks, your paper will be copyedited to ensure that it conforms to Nature Ecology and Evolution style. Once your paper is typeset, you will receive an email with a link to choose the appropriate publishing options for your paper and our Author Services team will be in touch regarding any additional information that may be required.

Due to the importance of these deadlines, we ask you please us know now whether you will be difficult to contact over the next

month. If this is the case, we ask you provide us with the contact information (email, phone and fax) of someone who will be able to check the proofs on your behalf, and who will be available to address any last-minute problems. Once your paper has been scheduled for online publication, the Nature press office will be in touch to confirm the details.

Acceptance of your manuscript is conditional on all authors' agreement with our publication policies (see www.nature.com/authors/policies/index.html). In particular your manuscript must not be published elsewhere and there must be no announcement of the work to any media outlet until the publication date (the day on which it is uploaded onto our web site).

Authors may need to take specific actions to achieve compliance with funder and institutional open access mandates. If your research is supported by a funder that requires immediate open access (e.g. according to [Plan S principles](https://www.springernature.com/gp/open-science/plan-s-compliance) or the [NIH public access policy](https://www.springernature.com/gp/open-science/us-federal-agency-compliance)) then you should select the gold OA route, and we will direct you to the compliant route where possible. Because authors warrant under our subscription licensing terms that they haven't committed to licensing any version of their article under a licence inconsistent with the terms of our agreement – including the applicable embargo period – publication under the subscription model isn't suitable for authors whose funders require no embargo.

We welcome the submission of potential cover material (including a short caption of around 40 words) related to your manuscript; suggestions should be sent to Nature Ecology & Evolution as electronic files (the image should be 300 dpi at 210 x 297 mm in either TIFF or JPEG format). Please note that such pictures should be selected more for their aesthetic appeal than for their scientific content, and that colour images work better than black and white or grayscale images. Please do not try to design a cover with the Nature Ecology & Evolution logo etc., and please do not submit composites of images related to your work. I am sure you will understand that we cannot make any promise as to whether any of your suggestions might be selected for the cover of the journal.

You can generate the link yourself when you receive your article DOI by entering it here: <http://authors.springernature.com/share>.

[redacted]

P.S. Click on the following link if you would like to recommend Nature Ecology & Evolution to your librarian <http://www.nature.com/subscriptions/recommend.html#forms>

** Visit the Springer Nature Editorial and Publishing website at http://editorial-jobs.springernature.com?utm_source=ejP_NEcoE_email&utm_medium=ejP_NEcoE_email&utm_campaign=ejp_NEcoE for more information about our career opportunities. If you have any questions please click [here](mailto:editorial.publishing.jobs@springernature.com).

Reviewer 1:

The authors investigated the morphology and carbon isotopic composition of lycopphyte sporophylls across the Permian-Triassic mass extinction event and found evidence that lycopphytes during and shortly after the extinction were particularly adapted to hot-house conditions. This proposition is supported by a climate model that reconstructs paleotemperatures in the studied habitats. The authors conclude that this physiological adaptation may have enabled the terrestrial biosphere to survive the prolonged hot-house event.

I emphasize that I am not a paleobotanist and can therefore not comment on those aspects of the manuscript. My comments will focus on aspects of isotope geochemistry and Earth system science.

I find the results overall intriguing and impressive. The carbon isotope study is done carefully, with appropriate comparison to background biomass and other plants within the same sample. The interpretation of this dataset as evidence of CAM-like photosynthesis is compelling (although I stress that I cannot comment on the plausibility of CAM photosynthesis in these particular organisms at this time, because that is outside of my field). To an outsider, the paleobotanic dataset looks convincing. However, I wondered if there is any risk that the conclusions may be skewed by the fact that the new morphology-based taxonomy lacks resolution (as noted in the manuscript). Is there any risk that ecophysiological information could be misinterpreted (see also comment below)?

Response: We appreciate the reviewer pointing out these risks and giving us valuable suggestions. We address these issues throughout the response letter as they are raised.

Regarding the overall conclusions about broader implications for survival of the biosphere and feedbacks on global climate, it would be helpful if information could be provided on what fraction of the terrestrial flora was composed of CAM-photosynthesizers at any given time. The discussion states that these were dominant and controlling carbon burial globally, but the $\delta^{13}\text{C}$ data imply that other non-CAM plants were present. A mass balance calculation would be helpful to quantitatively support this broader conclusion and the title of the manuscript.

Responses: We appreciate this suggestion and have considered how this might be done. Unfortunately, the current plant fossil record does not allow for reliable biomass estimates because of large spatial gaps in preservation and a direct inversion of the $\delta^{13}\text{C}$ record involves a large number of uncertainties in the compositions and fluxes of other isotopically enriched or depleted sources. We believe that the best way to assess this would be through a new vegetation model that explicitly includes a CAM photosynthesis module—something lacking in most existing vegetation models. This will allow us to quantify global C_3 and CAM vegetation distributions and biomass proportions based on fossil plant functional traits and to place this within a wider Earth system model that can output $\delta^{13}\text{C}$ records for testing. We value this suggestion and aim to pursue it in future work.

Line comments:

I. 31 and 49: the word ‘dramatic’ has no scientific meaning. You can delete it without loss of content.

Response [Line 31, 49]: Deleted.

I. 116: data are

Response [Line 121]: Modified.

II. 231-244: I appreciate that the authors acknowledge the limitations of their approach. However, it leaves me wondering if the loss of taxonomic resolution could lead to false interpretations about the ecophysiology of these organisms. In other words, could sporophytes from organisms adapted to distinct environmental conditions still look identical, based purely on morphological characteristics? And so, could it happen that these fossils are misinterpreted?

Response: We agree that morphology alone cannot always capture ecophysiological divergence, particularly in closely related taxa or under convergent environmental pressures. However, modern *Isoetes* species generally occupy broadly similar semi-aquatic to aquatic habitats with overlapping environmental tolerances, which reduces the likelihood that morphologically similar taxa in our fossil dataset would have occupied radically different ecological niches. Furthermore, the Lepidodendrales lineage shows strong evolutionary conservatism in both morphology and habitat preference, with structural traits traceable to their Carboniferous relatives. Some of these Palaeozoic forms may already have possessed anatomical preconditions for CAM photosynthesis, making it plausible that this physiological capability persisted over geological timescales. These factors, combined with the fact that our principal ecophysiological interpretations are made at the genus/clade level and supported by independent $\delta^{13}\text{C}$ evidence and climate modelling, suggest that any species-level misclassification is unlikely to systematically bias our broader ecological conclusions. We have clarified this in the Discussion [Line 387-390].

LI. 252-254: coastal lowlands can be very heterogeneous in salinity, as they are located along the interface between freshwater and seawater. Can additional evidence be provided that salinity is not a concern for this analysis?

Response: We appreciate the reviewer’s comment and have carefully considered the potential influence of salinity. To test this, we compared lycophyte and non-lycophyte samples collected from the same stratigraphic layers and localities (e.g., lycophyte sample 1-3-1, 1-5-1, 1-6-1, 1-7-1, 1-7-1, 1-8-1, 1-20-1, 1-21-1, 1-14-1, 1-12-1, non-lycophyte sample, 1-10-1, 1-11-1, 1-16-1, 1-19-1, 1-17-1, 1-17-1, 1-15-1, in Supplementary Fig. S6). We also compared the same lycophyte and non-lycophyte species across different localities and coastal depositional facies (e.g., lycophyte *Lepacyclotes* samples 1-8-1, 1-9-1, 1-20-1, 1-21-1, 1-14-

1, 1-12-1; non-lycophyte *Neocalamites* samples 6-1-4, 1-11-1, 4-4-1). We also plot the $\delta^{13}\text{C}_{\text{org}}$ by sedimentary facies for late Permian to Middle Triassic lycophyte and non-lycophyte plants in the supplementary file 3 [Figure S8]. In figure S8, depositional facies do not appear to be the primary control on plant $\delta^{13}\text{C}_{\text{org}}$. The four facies (Marine, Lagoon, Delta & coastal plain, and Fluvial/Lacustrine) show broad overlap along the $\delta^{13}\text{C}_{\text{org}}$ axis with no clear facies-specific banding. By contrast, within any given facies the data symbols coded by age separate out more consistently, and the color-coded genera show coherent centres and offsets from one another. Visually, this pattern implies that temporal background (e.g., atmospheric CO_2 – driven baseline shifts) and taxonomic composition exert stronger influence on $\delta^{13}\text{C}_{\text{org}}$ than facies. These results indicate depositional environment (including changes in salinity) was not the primary control on $\delta^{13}\text{C}_{\text{org}}$. Moreover, given the likely reduced reliance of lycophytes on stomatal CO_2 uptake due to the possibility of absorbing CO_2 through their roots and the low stomata density of cuticle fossils, theoretical expectations are that salinity would exert only a minimal effect on their isotope values.

Furthermore, to address broader concerns about taphonomy and preservation on the taxonomic analysis as reviewer 3 pointed out, we added a supplementary PCA including morphometric data against sedimentary facies [Figure S7 in supplementary file 3]. This was done solely for the South China taxa given that we completed a detailed sedimentary facies identification for them. This shows Triassic lycophyte genera occupy distinct regions of the morphospace, and samples of the same genus from different sedimentary facies consistently cluster within the same region. This pattern highlights the strong taxonomic signal and morphological conservatism of these genera, indicating that taphonomic processes were not the primary factor influencing taxonomic differentiation.

Both analyses confirm that neither taphonomy nor environmental factors, including salinity, significantly influence our results. We added a new sentence in the manuscript to emphasize that [Line 255-266].

l. 269: delete the comma

Response [Line 281]: Deleted.

l. 274: data indicate (the word ‘data’ is plural)

Response [Line 285]: Thanks, deleted.

l. 277: change to “...share a similar carbon isotope composition of circa...”

Response [Line 288]: Modified.

l. 409: Maybe specify what the non-lycophytes are, so that the reader can be more certain that they could not have been performing CAM-based photosynthesis. I am not familiar with how widespread this pathway is among plants and if it could be present in non-lycophytes of that age. Other readers may have the same question.

Response [Line 302-303, 436-438]: Good point. We have added this.

l. 446: is ‘dominated by CAM plants’ the correct phrase to use? After all, there are still many non-lycophytes in the samples that do not express CAM photosynthesis (as evidenced by the isotopic data).

Response: By “dominated by CAM plants”, we are referring specifically to the macrofossil record, which shows lycophytes as the pioneer taxa widely reported across post-PTME terrestrial and paralic strata at all latitudes, as outlined in the Introduction. However, we acknowledge that plant macrofossils provide only a partial record and cannot capture “hidden floras” from upland regions. Published palynological data indicate upland survivors worldwide, and the bulk terrestrial organic carbon isotope signal likely integrates inputs from both lowland and upland vegetation, as well as phytoplankton and microorganisms—components not fully represented in the macrofossil record. Moreover, although CAM plants may have been dominant in abundance, their small size and biomass suggest that they were not necessarily the main contributors to the bulk-rock carbon isotope signal. We agree this is an important direction for further work, and we have revised the conclusions [Lines 495–500] to explicitly highlight the need for future carbon isotope modelling that integrates vegetation dynamics with CAM function.

l. 445: related to my previous comment: Terrestrial carbon burial would only change significantly if CAM plants made up a large fraction of the terrestrial flora. This point is important to resolve. Perhaps an isotopic mass balance calculation could be applied to the samples to determine what fraction of the terrestrial biomass was CAM-derived?

Response [Lines 495–500]: We are very eager to perform a carbon isotope mass-balance analysis for terrestrial vegetation. But to underscore the difficulty in doing this, we cite our recent study published during the review process (<https://doi.org/10.1038/s41467-025-60396-y>) which uses an Earth system model to simulate the PTME carbon cycle, and demonstrates how carbon isotopes are affected by a combination of biotic and abiotic processes over this timeframe. With our new findings on CAM, we will be able to improve this model – which did not include CAM – to test how the rise of CAM populations affected the carbon cycle and carbon isotopes, but we believe that this requires a follow-on publication.

l. 462: Also here, it needs to be shown first that these organisms were indeed the major remnants, in comparison to other plants.

Response [Lines 477]: We have added citations here to support this point.

ll. 462-465: This sentence is not entirely clear. Why would the absence of CAM plants have extended the hothouse conditions beyond 5 million years? This point seems to contradict the statement in ll. 454-456, which mentioned a positive feedback that increased warming. Please clarify.

Response [Lines 501-507]: We appreciate the reviewer's careful reading. We agree that our original wording was ambiguous and have now clarified this point in the revised text. Our intention was to highlight two complementary but distinct aspects of CAM lycophyte survival: 1. CAM-dominated floras had lower productivity and weaker bio-weathering capacity than the pre-extinction forests, which reduced terrestrial carbon burial and nutrient fluxes to the ocean. This acted as positive feedback that amplified Early Triassic warming; 2. Without CAM-capable lycophytes persisting as pioneer taxa, terrestrial ecosystems may have suffered an even more complete collapse, leaving vast areas barren. Such a scenario could have further slowed the eventual recovery of biogeochemical cycles and prolonged the super-greenhouse conditions far beyond ~5 Myr. The scenarios we describe are consistent with the recent modelling study by Rogger et al. (2024, *Science*), which tested different assumptions about plant thermal sensitivity and ecosystem persistence.

Fig. 5: It would be helpful to add ages in numbers to make this figure easier to read for people who are not familiar with the stage names. It would make it easier to compare to Fig. 4.

Response [Figure 5]: Thanks for the suggestion. As terrestrial strata are difficult to correlate precisely with the marine GSSP, the Permian–Triassic Boundary on land is commonly referred to as the Permian–Triassic transition. Our simulations for the latest Permian (Changhsingian) and earliest Triassic (Induan) both fall very close to this boundary; therefore, we indicate only the approximate timing of the transition in Figure 5. The stage names in Figure 5 are provided to facilitate direct comparison with Figure 4.

II. 676-679: Please provide information about analytical reference materials and report precision and accuracy of the data.

Response [Line 727-746]: Thanks for pointing this out. We have now added the information on analytical reference materials, precision, and accuracy, as reported in the standard laboratory reports.

Reviewer 2:

This is a thought-provoking study. Xu et al. use lycophyte sporophyll morphology to assemble a new phylogeny and argue that the highly prevalent lycophytes in the aftermath of the Permian-Triassic mass extinction had a survival advantage due to employing CAM photosynthesis. They bolster the interpretations of their morphological trait-based phylogeny with carbon isotope data from fossil lycophytes showing diminished carbon isotope fractionation (consistent with CAM) during the PTME, as well as climate model simulations showing prohibitively high temperatures for C3 photosynthesis.

Overall I think this is a neat set of observations that is tied together with a plausible explanation. The field would benefit from seeing this in the literature. That said, many details are not (and cannot be) fully sorted out. I think the paper works well as a “hypothesis paper”,

rather than a definitive assessment of the photosynthetic metabolism of fossil lycophytes and its role in lycophyte (or broader biosphere) survival (to that end, my first suggestion is to end the title with a question mark!). I have some suggestions below to open up a bit more discussion on points of contention, and with attention to those I think this would be acceptable for publication.

Response: We thank the reviewer for their constructive suggestions and thoughtful perspective. We have revised the text accordingly (with details below), clarifying points of uncertainty and expanding the discussion where data cannot provide a definitive resolution.

I am less familiar with the basis for the morphological trait analyses, so my comments pertain to the carbon isotopes and plant physiology:

1. Carbon isotopes and the inference of CAM. While C isotopes are a crucial piece of evidence for inferring CAM activity in deep time, they alone can't be definitive, for a few reasons. First, the C isotope effects resulting from CAM overlap the lower range of those generated by C₃ photosynthesis. In terms of carbon mass balance, that end-member of C₃ photosynthesis is similar to CAM (being more efficient at C fixation). But metabolically, inferring CAM means the plants are employing organic acid accumulation with temporal separation of C uptake & fixation, whereas the low-D₁₃C end-member of C₃ photosynthesis does not. Second, we don't precisely know the D₁₃C values, because we don't precisely know the d₁₃C_CO₂ value through the event. So the absolute value of D₁₃C cannot be used to distinguish CAM vs. C₃. In light of this, the authors compare d₁₃C of lycophytes to non-lycophytes through the PTME. This is sensible, but non-lycophytes sample sizes are small, particularly for the "transition" phase. Third, even if the n=3 sampling of non-lycophytes captures an accurate value for the "transition" phase, it is possible that these outgroup plants respond differently than lycophytes to CO₂, moisture or insolation, such that both groups moved to lower d₁₃C due to the drop in d₁₃C_CO₂, but the non-lycophytes increased their D₁₃C (e.g., due to high CO₂) while lycophytes remained the same. In such a scenario, there would be no reason to infer CAM in the lycophytes. As a final note, I appreciate the discussion of aqueous CO₂ uptake, but I don't think the isotopic data can give a clear vote for or against it (very uncertain what the d₁₃C of that CO₂ source would be, depending on amount of sedimentary respired carbon vs. marine DIC). I mention these issues not necessarily to say I think it is unlikely that the lycophytes employed CAM, but rather to say the available data are not definitive.

Response: We thank the reviewer for this thoughtful and constructive critique. We fully agree that these are important limitations that should be further discussed. To underscore the need for further investigation here we have revised the text to frame CAM as the most parsimonious explanation given the available evidence, but not to close the question definitively.

Specifically, we interpreted the isotopic data in combination with: (1) the phylogenetic placement of the studied lycophytes within Isoetales, which include modern representatives with demonstrated CAM or CAM-like physiology [**Lines 216–217, 375–381**]; (2)

morphological traits of Triassic lycophytes (e.g., aerenchyma and root-based CO₂ uptake) that are consistent with CAM functioning [Lines 381–390]; and (3) the paleoenvironmental context of extreme seasonal drought and high evaporative stress, where CAM physiology would confer a survival advantage [Lines 339–355].

We also acknowledge the uncertainty introduced by small non-lycophyte sample sizes. We were able to analyse all four non-lycophyte specimens available to us, but the extreme climatic stress of the Permian–Triassic transition severely limited plant survival and fossil preservation [Lines 458–460]. Moreover, these post-PTME non-lycophyte remains are very small in size (commonly 2–5 mm²), making it necessary to combine multiple fragments to obtain reliable isotopic measurements. We now explicitly highlight this limitation in the revised manuscript [Lines 267–278].

Finally, we agree with the reviewer that aqueous CO₂ uptake cannot be resolved from isotopic data alone. We acknowledge this limitation [Lines 311–316] and adopted a more balanced tone around this aspect of the work.

2. Temperature limits of extinct C₃ plants and CAM as a survival mechanism. The statement in Line 330: “The survival of these plants under such extreme heat [45-65 C] strongly suggests the use of an alternative photosynthetic pathway, rather than the C₃ type.” needs more unpacking. First, while 45-65C is indeed terribly hot, the cited paper (ref. 19) shows that CAM plants have lower optimal growth temperatures than C₃, with only C₄ having higher temperature tolerance. On this basis, it would seem that CAM does not help explain lycophyte survival. Second, as noted in Line 322, extant plant lineages (not just C₃, but CAM as well) evolved long after the interval being studied here. The applicability of modern experiments (as in ref. 19) to these ancestral lineages is difficult to prove. There’s no way around that, but it remains an inherent limitation of this sort of study. Third, if higher temperature did indeed favor CAM, please explain the mechanism in more metabolic detail. The additional sentences about lycophyte survival in the Discussion (lines 459-462) don’t add much to that, and also bring in tangential ideas such as “anti-heavy metal antioxidant enzyme systems” that are not clearly tied to the environmental perturbations being discussed. Overall, the role of CAM in lycophyte survival needs better explaining.

Response: We thank the reviewer for raising this important point. We have added the appropriate reference to clarify that while CAM plants do not necessarily exhibit higher thermal optima than C₃ or C₄ plants, they can tolerate temperatures up to ~70 °C—substantially higher than the limits typically observed for C₃ plants [Line 339-355]. CAM confers a survival advantage under the combined stresses of extreme heat and aridity that characterized the Early Triassic. In such conditions, the critical limitation for C₃ plants is balancing water loss and photorespiration, which can lead to lethal conditions when their stomata remain closed at high temperatures to minimize water loss through evapotranspiration. In contrast, CAM plants open their stomata at night to fix CO₂ into organic acids, then close them during the hot part of the day, thereby minimizing water loss and avoiding excessive photorespiration [Line 391-398]. This strategy is particularly

advantageous in a “mega–El Niño world” of the Permo-Triassic Transition with prolonged seasonal droughts and extreme heat.

The additional sentences in the Discussion were intended to highlight that: (1) CAM plants possess other mechanisms to cope with extreme conditions, such as entering dormancy; and (2) beyond high temperatures, the Early Triassic was also marked by heavy metal stress from Siberian Traps volcanism and strong seasonality. These stressors align with the typical physiological traits of Triassic lycophytes and their modern relatives (Isoetales), reinforcing the idea of ecological and genetic continuity between fossil and extant lineages. We modified these sentences to make these ideas clearer [Line 461-473].

Reviewer 3:

Zhen Xu et al present an important, data rich and multi-faceted analysis on the taxonomy, ecophysiology and climatic / biogeographic distribution of fossil herbaceous lycophytes from before, during and after the Permian-Triassic mass extinction event. The study brings fresh insights, new analyses and a multi-disciplinary approach to furthering understanding of plant species resilience to extreme climatic warmth during this time interval. Xu et al propose based on detailed morphological character analysis of 485 fossil lycophyte specimens that the Permian-Triassic transition interval is characterised by a taxon of herbaceous lycophytes which are morphologically (and taxonomically) and ecophysiological distinct from their ancestors and descendant clades. The author team argue that the P-T transition fossil likely possessed CAM photosynthesis similar to the living quillworts (Isoetids) based on having less negative isotopic value compared with contemporaneous non-lycophyte taxa and shared morphological traits. They go on to argue that CAM photosynthesis which has a higher daytime photosynthetic temperature tolerance than C3 taxa and much higher water use efficiencies would have conferred resilience to the surviving herbaceous lycophytes in the transition interval.

Overall the study is exciting, novel and robust and brings a wealth of new fossil data and analyses to the Permian-Triassic mass extinction interval. Although the idea of flexible photosynthesis as a strategy to withstand earth extreme events is not new in itself (see papers by Looe, Vischer, Green etc- all of which are cited in the manuscript) – the careful and detailed analysis of Lycophyte sporophylls as a primary source of data is highly novel.

Response: We are very grateful to Reviewer 3 for the thoughtful encouragement and the thorough, constructive suggestions, which have been of great help to us. We carefully reviewed both the summary points and the detailed comments in the annotated PDF, and we have revised the manuscript accordingly. In several places, we also incorporated additional analyses to address the reviewer’s concerns and to further strengthen the study. We believe these revisions have improved the clarity and robustness of our work, and we thank the reviewer for helping us achieve this. Detailed responses are provided below.

I think the paper can be further strengthened with a few additional considerations by the

author team as follows:

(1) Throughout the manuscript the authors talk about switching from C3 to CAM. The overall suggestion throughout the manuscript is that the PT transition taxa are obligate CAM? yet it is not stated if the authors think they are obligate or facultative CAM. Looking at the carbon isotope data I think it is equally parsimonious to suggest that the plants are facultative CAM – they carry out C3 photosynthesis predominantly but can switch within their lifetime to CAM photosynthesis under extreme conditions (eg submergence or high temperature/ high aridity stress). A subtle tightening up of the language should resolve this uncertainty.

Response: We agree that that CAM could have indeed been facultative and have modified the text in several places to make this clearer [Line 379, 391, 397, 400–402, 427, 443, 456–458, 479]. Many thanks for the note on this.

(2) Taphonomy and differences in depositional environment are somewhat glossed over in the PCA analysis. Could an additional set of analyses be included which codes the fossil taxa in the PCA morphospace with taphonomic or depositional environment/ facies ? This would help to test if the groupings within the PCA are indeed due to morphological similarities and differences due to evolutionary differences rather than local environmental signals? I don't think there is an issue here but it would be nice to conduct this test if it is possible.

Response: We appreciate this suggestion and have made two new figures in the supplementary file 3:

To address broader concerns about taphonomy and preservation on the taxonomy, we added a supplementary PCA including morphometric data against sedimentary facies [Figure S7]. We only did this for South China taxa because we completed a detail sedimentary facies identification for them. In figure S7, Triassic lycophyte genera occupy distinct regions of the morphospace, and samples of the same genus from different sedimentary facies consistently cluster within the same region. This pattern highlights the strong taxonomic signal and morphological conservatism of these genera, indicating that taphonomic processes were not the primary factor influencing taxonomic differentiation.

To evaluate the influence of growth and depositional environment on carbon isotope fractionation, we made Figure S8 which plots $\delta^{13}\text{C}_{\text{org}}$ against sedimentary facies for late Permian to Middle Triassic lycophyte and non-lycophyte plants. Figure 8 illustrates that depositional facies do not appear to be the primary control on plant $\delta^{13}\text{C}_{\text{org}}$. The four facies (Marine, Lagoon, Delta & coastal plain, and Fluvial/Lacustrine) show broad overlap along the $\delta^{13}\text{C}_{\text{org}}$ axis with no clear facies-specific banding. By contrast, within any given facies the data symbols coded by age separate out more consistently, and the color-coded genera show coherent centres and offsets. Visually, this pattern implies that temporal background (e.g., atmospheric CO_2 -driven baseline shifts) and taxonomic composition exert stronger influence on $\delta^{13}\text{C}_{\text{org}}$ than facies. These results indicate that depositional environment is not the primary control on $\delta^{13}\text{C}_{\text{org}}$. Moreover, given the likely reduced reliance of lycophytes on stomatal CO_2 uptake due to the possibility of absorbing CO_2 through their roots and the low

stomata density of cuticle fossils, theoretical expectations are that salinity would exert only a minimal effect on their isotope values.

Both analyses confirm that neither taphonomy nor environmental factors, including salinity, significantly influence our results. We added a new sentence to emphasize this [Line 260-266].

(3) There seems to be a small inconsistency in the CO₂ trends described in the intro and those used for the HAD gcm analyses – please double check.

Response: We modified the main text to the right number from the reference and have added an explanation in the Methods section [Lines 70, 786–790], clarifying that we used the maximum boundary CO₂ value derived from the combined constraints of plant stomata, paleosols, and carbon isotope proxies for the simulation.

(4) I have made detailed comments and suggestions throughout the attached manuscript file with all my additional minor comments to improve the manuscript.

Response: Many thanks! We have checked these comments one by one and modified the manuscript to address them each. The modified parts are highlighted in blue in the final documents. Our response to some of the comments follow:

1. References modified;
2. Line 148-153 characters are not listed in the order of the loading score due to the different values on the two axes, and we added a sentence in Line 98-100 that we do have at least one specimen of the whole plant for each genus that identifies the whole plant growth habitat; pictures are provided in supplementary file 3;
3. We sadly don't have conifer macrofossils at the Permian-Triassic transition, and only have the seed fern *Germaropteris* (*Lepidopteris*, *Peltaspermum*) and lycophyte *Tomiostrabus* to compare at the PTT. This absence reflects the very limited macrofossil records for the key extinction interval;
4. We have nighttime temperature simulations from the model, which we hope to employ for the next stage of our work: incorporating CAM into vegetation models to better simulate deep-time plant dynamics;
5. Although our results suggest that *Tomiostrabus* at the Permian–Triassic transition likely employed facultative CAM, we cannot exclude the possibility that other lycophytes also possessed this capacity. However, under the less stressful environmental conditions of their time, CAM expression may not have been necessary and thus left no detectable signal. For example, Permian *Lepidodendron* had abundant aerenchyma tissues similar to modern *Isoetes*, and Looy et al. (2021) hypothesized that the Late Triassic Pleuromiaceae *Mesenteriophyllum* may likewise have been capable of CAM.

In summary, this is a novel, exciting and data rich paper that combines taxonomic , phylogenetic, paleoecophysiological and paleogeographic and paleoclimate analyses to make a strong case that the disaster herbaceous taxa which were present during the Permian-Triassic transition interval are phylogenetically distinct from their Permian ancestors and possessed novel photosynthetic biology that enabled them to survive one of the greatest extinction events in Earth history.

Response: We thank the reviewer again for the positive assessment and constructive suggestions.

Reviewer 1: (Remarks on code availability):

The code is very complex and beyond my area of expertise. However, it is based on previously published modelling frameworks. I therefore trust that it is reliable.

Response: We thank the reviewer for their comment. The modelling framework used in this study is based on previously published and peer-reviewed models, which are appropriately cited in the manuscript. All newly developed code used for this study has been provided in the Supplementary Information, ensuring that the results are fully transparent and reproducible.

Reviewer 2:

The authors have adequately addressed my comments and those of the other reviewers. I am happy to see this version of the manuscript published.

Response: We thank the reviewer for their constructive comments and are pleased that they find the revised manuscript suitable for publication.

Reviewer 3:

It is a pleasure to re-evaluate this manuscript on 'Cam photosynthesis, a key trait in surviving Earth's largest extinction'. I have read the detailed responses to reviewers and the revised manuscript and feel that the authors have thoroughly addressed my initial concerns and those of the other reviewers. The tone of the manuscript is very well pitched following the reviewer comments highlighting the advances the paper makes in relation to the evidence for CAM and also avenues for further testing of the CAM hypothesis. I think this is a really important paper and makes a significant advance to our understanding of how vegetation can withstand extreme intervals in Earth history. It also elegantly demonstrates the importance of paleobiology in understanding past and current earth system processes which are becoming increasingly relevant as the Earth transitions from a coldhouse to coolhouse climate state.

I have one remaining minor suggestion:

Line 351 the authors state that 'Alternatively, CAM photosynthesis has been previously hypothesized in deeper time^{12,38,47,48}

Please add two further citations to support this statement as both have suggested earlier evolution of CAM and ways of detecting it in the fossil record.

Raven JA, Spicer RA. The evolution of crassulacean acid metabolism. In Crassulacean acid metabolism: biochemistry, ecophysiology and evolution 1996 (pp. 360-385). Berlin, Heidelberg: Springer Berlin Heidelberg.

McElwain JC, Mattheaus WJ, Barbosa C, Chondrogiannis C, O'Dea K, Jackson B, Knetge

AB, Kwasniewska K, Nair R, White JD, Wilson JP. Functional traits of fossil plants. *New Phytologist*. 2024 Apr;242(2):392-423.

Response: We thank the reviewer for their detailed, thoughtful, and highly encouraging assessment of our revised manuscript. We are pleased that the revisions have addressed their concerns and those of the other reviewers. We agree that the suggested references are highly relevant, and both have now been added to the manuscript to support this statement [**Line 308**].